# OD-Stega: LLM-Based Near-Imperceptible Steganography via Optimized Distributions

## Abstract

We consider coverless steganography where a Large Language Model (LLM) drives an arithmetic coding decoder to generate stego-texts. An efficient method should embed secret message bits in as few language tokens as possible, while still keeping the stego-text natural and fluent. We show that on the individual token level, this problem is mathematically equivalent to maximizing the entropy of a replacement probability distribution of the next token generation, subject to a constraint on the KL divergence between the chosen probability distribution and the original distribution given by the LLM. A closed-form solution is provided for the optimization problem, which can be computed efficiently. Several important practical issues are also tackled: 1) An often-overlooked tokenization mismatch issue is resolved with a simple prompt selection approach, 2) The combination of the optimized distribution and the vocabulary truncation technique is considered, and 3) The combination of the optimized distribution with other sequence-level selection heuristics to further enhance the efficiency and reliability is studied.

## 1 Introduction

In a steganography system, Alice, the sender, aims to convey a secret message to Bob, the receiver. The carrier signal can take the form of text, image, audio, or video (Anderson & Petitcolas, 1998; Cox et al., 2007; Provos & Honeyman, 2003). In this work, we focus on natural language text messages as the type of carrier signals, and in this case, the resultant signal with the secret message embedded is referred to as the stego-text. Alice transmits the stego-text to Bob via a public channel, which is being monitored by an eavesdropper Eve. Eve wishes to determine whether there is a hidden message in the stego-text. Alice must ensure that the stego-text can be decoded correctly by Bob, and at the same time, guarantee with a high probability that Eve cannot detect whether a secret message exists or not. A good analogy is that Bob is a prisoner, Alice is the family member outside the prison who has a letter for Bob, and Eve is the prison guard who may confiscate the letter if something unusual is detected about the letter (Simmons, 1984).

Conventionally, steganography relies on an existing cover signal (cover text), and achieves steganography by making subtle changes imperceptible to Eve on the cover text. For example, Alice can replace certain words by their synonyms following pre-agreed patterns (Topkara et al., 2006; Chang & Clark, 2010; Safaka et al., 2016). Recently, as generative models, particularly large language models, become more and more powerful, coverless steganography has shown significant performance advantages. With this approach, the stego-text appears indistinguishable from natural languages, and more importantly, a large amount of the secret information can be hidden in shorter stego-texts than the traditional cover-text-based approaches (Fang et al., 2017; Yang et al., 2018; Ziegler et al., 2019; Xiang et al., 2017; Dai & Cai, 2019; Zhang et al., 2021; Shen et al., 2020; Kaptchuk et al., 2021; Ding et al., 2023; de Witt et al., 2024).

The underlying driver for LLM-based steganography is usually the arithmetic coding (AC) algorithm (Witten et al., 1987), which is an efficient data compression algorithm based on the idea that any finite-length finite-alphabet data sequence (e.g., text) can be mapped to a small interval in the range of $[0, 1)$ based on the cumulative probability distribution function. Therefore, a binary representation that accurately specifies this interval is a compressed representation of the sequence. The decompression process can reverse this encoding process and recover the original sequence. In LLM-based steganography, Alice utilizes the arithmetic coding **decoder**, together with the probability distribution produced by LLM, to map the secret binary sequence to a stego-text. Bob can then recover the

secret message by performing the arithmetic encoding, which is assumed to have access to the same LLM. Intuitively, the arithmetic coding decoder essentially performs token-wise sampling following the conditional probability distribution given by the LLM, using the secret message bits as the starting randomness (from a uniform distribution to a non-uniform one). The stego-text would appear natural and fluent if the LLM captures accurately the true distribution of natural languages.

In many use scenarios, the security requirement in steganography can in fact be relaxed: 1) when Eve is computation-bounded (e.g., in a mobile device), 2) when Eve is delay-constrained (e.g., in streaming processing or time-sensitive applications), or 3) under societal constraint (e.g., censorship under constitutional right protection). In such cases, Eve can be modeled as a weak detector, and correspondingly the steganography security requirement can be relaxed to take advantage of the situation. This consideration is in fact already implicit in several previous works invoking "near-imperceptibility" (Dai & Cai, 2019; Shen et al., 2020) where the LLM next token probability distributions were truncated or positions were skipped to either reduce computation or avoid excessive distribution mismatch. Clearly, the "perceptibility" of a casual user is different from that of an expert, and the authors there used the KL divergence to quantity the security loss, and studied its relation with the embedding capability when more truncation is taken or more positioned are skipped. Further generalizing this idea, we can replace the conditional probability distribution produced with another distribution, as long as the replacement mechanism is deterministic and causal, such that Alice and Bob remain synchronized, when the steganography security requirement is less stringent.

Taking this generalized view, our work is based on the following observation. There appears to be a fundamental tradeoff between the amount of secret bits one can hide in the stego-text and the detectability of steganography; the former consideration is usually measured by the embedding capability or embedding utilization in the literature (Dai & Cai, 2019; Shen et al., 2020; Kaptchuk et al., 2021; Ding et al., 2023). Improving the utilization is particularly important for LLM-based steganography, since the generative process in LLMs can become almost deterministic, and it becomes difficult to embed secret bits unless a very long stego-text is used. In the context of LLM-based steganography and the underlying arithmetic coding decoder, this requirement at the token level essentially translates to maximizing the entropy of a replacement probability distribution, subject to a constraint on the distance between this new token generation distribution and the original one produced by the LLM. We formalize this problem under the KL divergence constraint, and show that it has a closed-form solution that can be computed efficiently. We refer to this steganography approach via an optimized distribution simply as OD-Stega.

Our formulation formalizes the general tradeoff between steganography security and embedding utilization, and our approaches can be specialized to previous methods. For example, in the perfectly secure extreme, our approach essentially reduces to the approach given in Kaptchuk et al. (2021). Moreover, given the fundamental nature of the mathematical formulation, our approach can also be straightforwardly incorporated into other methods such as Ding et al. (2023); Zhang et al. (2021).

In addition to the principled formulation outlined above, our work also tackles several practical issues. Firstly, previous works using LLM for steganography assumed that the tokenizer is one-to-one, such that Bob can decode correctly every time. However, modern tokenizers in LLM are often not one-to-one, and therefore, these approaches often encounter decoding errors. We propose a simple strategy to remedy this issue through LLM prompting selection. Secondly, we combine OD-Stega with the existing technique of vocabulary truncation to reduce the computation complexity, and analyze the overall KL divergence of this strategy. Lastly, we combine the proposed single-token probability adjustment technique with other heuristics on the sequence level, and adaptively select optimization parameters based on the conditional entropy for each token. We conduct extensive experiments and demonstrate that the proposed approach can indeed embed significantly more secret message bits into the stego-text, and the generated stego-text remains perpetually indistinguishable.

The contribution of this work can be summarized as follows: 1) We provide a principled formulation to optimize the generative conditional distribution in order to embed more bits in shorter stego-texts; 2) We design an efficient algorithm to compute the optimized distribution for each token; and 3) We tackle several practical issues and provide strategies to combine OD-Stega with other methods to improve the efficiency and reliability at the sequence level.

We defer a detailed discussion on related works to the appendix.

## 2 PRELIMINARY

### 2.1 LLM-BASED STEGANOGRAPHY

A Large Language Model (LLM) can provide an estimate for the conditional probability distribution for the next token, given the sequence of tokens preceding it (Vaswani, 2017; Brown, 2020; Touvron et al., 2023). To generate a natural language sequence, one can sample the tokens from these distributions in an autoregressive manner. We next provide some notation for the rest of the paper.

Following the work of Shen et al. (2020) for LLM-based steganography, we assume that the secret message bit sequence $S$ is already encrypted, before Alice starts to embed it in the stego-text. Before encoding, Alice selects an initial prompt text $T_p$, independent of $S$, which typically determines the nature or semantic of the resulting stego-text. To encode $S$, Alice uses an encoding function $f(T_p, S)$ to produce a sequence of tokens $\underline{x}_{i>0} = (x_1, x_2, x_3, \ldots)$, which is then converted to the corresponding stego-text $T_s$ via detokenizing. The prompt and the stego-text $(T_p, T_s)$ are sent on the public channel. Bob first converts $T_s$ into the token form $\underline{x}_{i>0}$, then uses a decoding function $g(\cdot)$ such that $g(T_p, \underline{x}_{i>0}) = S$.

In LLM-based steganography, both $f$ and $g$ rely on the same LLM. At time $i$, an LLM takes the tokenized input $\underline{x}_{i-1} = (x_{-n_p-1}, x_{-n_p-2}, \cdots, x_{i-1})$ as the prompt, where $\underline{x}_0 = (x_{-n_p-1}, x_{-n_p-2}, \cdots, x_0)$ represents the tokenized sequence of $T_p$ and $n_p$ is the number of tokens in $T_p$. This produces the probability distribution $P_{LLM}$ for the next token $x_i$. We shall write it as $P_{LLM}(\mathbf{Y} = x_i \mid \underline{x}_{i-1})$, or simply $P^i$, which is the conditional probability for the next token, given the proceeding tokens (in the context window).

### 2.2 ARITHMETIC CODING

Several authors have shown that Arithmetic Coding, or AC for short, can be used together with language models to perform steganography (Ziegler et al., 2019; Shen et al., 2020; Ivasenko et al., 2021). AC is a method for data compression that encodes a whole sequence of symbols as a single value, based on the probability distribution. Typically, AC compresses the character in the sequence sequentially into a sequence of bits at the transmitter, and converts the sequence back to text during decompressing. The main idea of using AC for steganography is that an AC decoder can be viewed as a sampler in the set of natural language paragraphs using the secret message as a random seed, and since the secret message is uniformly distributed on the message set, the sampled text would look like natural language. Note that the AC encoding procedure is the steganography decoding procedure, and the AC decoding procedure is the steganography encoding procedure.

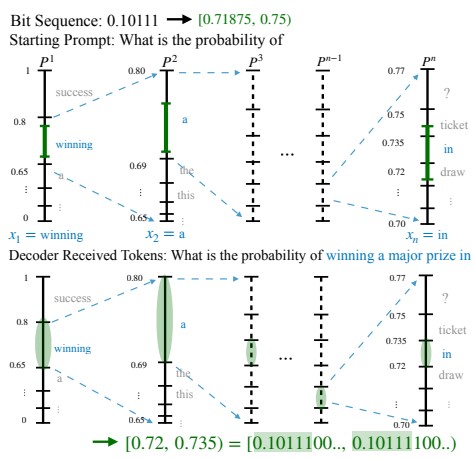

Figure 1: Example of AC in steganography

An illustrative example is given in Figure 1. Initially, a secret bit string is transformed into a decimal fraction interval. For instance, the sequence 10111 can be represented as the interval $\mathbf{I} = [0.101110000\cdots_2, 0.1011111111\cdots_2) \simeq [0.71875, 0.75)$. Next, we identify the range where this interval falls in the probability distribution $P^i$.

As illustrated in Figure 1, when we input the starting prompt "What is the probability of", the LLM generates a probability distribution for the most likely next tokens $P^1$. Based on this distribution, we determine where the interval lies. In this example, the interval corresponds to the token "winning", so we select the first token of the stego-text as $x_1 = $ "winning".

Once the first token $x_1$ is selected, the probability distribution $P^2$ is obtained by the same procedure of prompting $\underline{x}_1$ into the LLM. The next token $x_2$ is chosen based on where the interval lies within this distribution. This process is repeated iteratively until there is no ambiguity regarding where the interval $\mathbf{I}$ falls into. As Figure 1 illustrates, the interval is outside any $P^n$ token interval

range; thus, after choosing the $n$-th token, the stego-text generation is completed with a total of $n$ tokens, which can be converted directly into the stego-text.

During the decoding phase, Bob recognizes the starting token of the stego-text from the received text. Bob can then derive the identical distribution $P^1$ from the same LLM with the starting prompt text. With those stego-text tokens he receives, Bob is then able to retrieve the probabilities $P^{i>0}$ and reconstruct the bit sequence, continuing this process until every bit is recovered.

## 3 PROPOSED METHODOLOGY

A well-known fact in data compression is that the expected minimum number of bits to represent a message symbol following a probability $P$ is $H(P)$, i.e., the entropy of symbol (Cover & Thomas, 1991), and AC is one algorithm that can compress at a rate close to the minimum value. The same relation holds for LLM-based steganography using AC, in the sense that the expected number of secret message bits that can be embedded for a given token position-$i$ is the entropy of the conditional distribution $H(P^i)$. For example, if a token has a conditional distribution of $\{\frac{1}{4}, \frac{1}{4}, \frac{1}{4}, \frac{1}{4}\}$ on four possible token values, then 2 bits of secret message can be embedded in the stego-text.

It is obvious that a slight modification to the probability distribution from the true natural language distribution is nearly imperceptible to a human reader (weak detector), or even to a computer program for that matter. We can take advantage of such an opportunity to make the conditional distribution $P$ more amicable for embedding secret message bits, i.e., choose a different distribution $Q$ such that the entropy $H(Q)$ is larger. As long as $Q$ is kept close to $P$ under certain measure, we expect the generated stego-text to be nearly imperceptible, which leads us to the formulation given next.

### 3.1 PROBLEM FORMULATION: SAMPLING STRATEGY UNDER PERCEPTION CONSTRAINT

We formulate the following optimization problem for each token at time instance-$i$.

$$\max_{Q_j^i, \, \forall j \in [1:N_i]} \quad H(Q^i) = \sum_{j=1}^{N_i} -Q_j^i \log Q_j^i \tag{1}$$

$$\text{subject to} \quad D_{KL}(Q^i || P^i) = \sum_{j=1}^{N_i} Q_j^i \log \left( \frac{Q_j^i}{P_j^i} \right) \leq \delta \tag{2}$$

$$Q_j^i \geq 0, \quad \forall j \in [1 : N_i] \tag{3}$$

$$\sum_{j=1}^{N_i} Q_j^i = 1 \tag{4}$$

$$Q_j^i = 0 \quad \forall j \in \mathbb{A}_i = [N_i + 1 : N] \tag{5}$$

Let $N = |\mathcal{V}|$ be the total number of symbols in the vocabulary. The objective function $H(Q^i)$ in (1) represents the standard Shannon entropy, where we use the logarithm of base 2, implying we will measure the information in bits. We seek to replace the natural language distribution probability distribution $P$ given by the LLMs with a new distribution $Q$ towards a larger entropy value, which usually means a more uniform distribution. This would allow for embedding a greater number of secret bits within a single token. It is crucial for the new distribution to be close to that of the natural language, which is ensured by the constraint in (2), that the divergence between $Q$ and $P$ does not exceed a small threshold $\delta$. Note the problem above is a convex optimization problem.

There are in fact various other metrics to quantify the difference between $P$ and $Q$, but we choose to use the KL divergence in this work, since it has a clear operational meaning and is well adopted in steganography, moreover, it is connected to the error exponent in hypothesis testing (Cover & Thomas, 1991).

Without loss of generality, we will assume throughout the rest of this paper that the elements in the vocabulary are already given in descending order according to the probabilities $P^i$. The set $\mathbb{A}_i$ in the constraint (5) corresponds to the index set of elements in the alphabets with zero probability,

i.e., $P_j^i = 0$. Clearly there is no need to adjust tokens with a zero probability, since otherwise, the resultant KL divergence will be unbounded; this consideration is reflected in (5). We denote the number of nonzero elements in $P^i$ as $N_i = N - |\mathbb{A}_i|$. As a result, the number of variables in this optimization problem is in fact $N_i$ instead of $N$.

## 3.2 The Optimal Probability Adjustment Strategy

The main theoretical contribution of the work is Theorem 1, which gives the solution to the optimization problem (1)-(5).

**Theorem 1** *An optimal probability solution $Q^i$ to the optimization problem (1)-(5) is given by*

$$Q_j^i = \begin{cases} \frac{P_j^{i\frac{u}{1+u}}}{\sum_{j=1}^{N_i} P_j^{i\frac{u}{1+u}}}, & \forall j \notin \mathbb{A}_i \\ 0, & \forall j \in \mathbb{A}_i \end{cases} \qquad (6)$$

*for some $u \geq 0$ when $\delta \in [0, \frac{1}{N_i} \sum_{j=1}^{N_i} \log(\frac{1}{N_i P_j^i})]$, otherwise*

$$Q_j^i = \begin{cases} \frac{1}{N_i}, & \forall j \notin \mathbb{A}_i \\ 0, & \forall j \in \mathbb{A}_i \end{cases}.$$

Observe that this solution adjusts each non-zero element's probability $P^i$ by an exponential factor in the range $[0, 1]$. In the extreme case of $u = 0$, the optimal $Q^i$ becomes a uniform distribution, resulting in a large KL divergence; on the other hand, for the extreme case of $u = \infty$, we obtain the original distribution, implying the KL divergence is zero. The following lemma provides a connection between the parameter $u$ and the divergence constraint $\delta$.

**Lemma 1** *For $Q_j^i = \frac{P_j^{i\frac{u}{1+u}}}{\sum_{j=1}^{N_i} P_j^{i\frac{u}{1+u}}}$ and any $\delta \in [0, \frac{1}{N_i} \sum_{j=1}^{N_i} \log(\frac{1}{N_i P_j^i})]$, there exists a positive $u$, such that the solution given in Theorem 1 satisfies the constraint (2) with equality*

$$D_{KL}(Q^i || P^i) = \sum_{j=1}^{N_i} Q_j^i \log\left(\frac{Q_j^i}{P_j^i}\right) = \delta.$$

The proofs of Theorem 1 and Lemma 1 are given in the appendix, which are obtained by a careful analysis of the KKT conditions. Note that the specified $\delta$ only places a meaning constraint within the range given in Theorem 1. Otherwise, the KL constraint is essentially too loose, and the optimal solution $Q^i$ defaults to a uniform distribution. It remains to solve for the value of $u$ that satisfies the KL constraint with equality. For this purpose, we establish the following lemma.

**Lemma 2** *For the assignment $Q_j^i = \frac{P_j^{i\frac{u}{1+u}}}{\sum_{j=1}^{N_i} P_j^{i\frac{u}{1+u}}}$, $i = 1, 2, \ldots, N_i$, $D_{KL}(Q^i || P^i)$ is monotonically non-increasing with respect to $u$ in the range $u \geq 0$.*

The proof of Lemma 2 is provided also in the appendix. This lemma demonstrates that the KL divergence decreases as $u$ increases. This property is particularly useful in finding the value of $u$ since we can easily determine $u$ numerically using a simple and efficient bisection search.

As an illustrative example, consider a probability distribution of four tokens with values $P^i = [0.4, 0.3, 0.2, 0.1]$ and a small $\delta = 0.0384$. Since this $\delta$ value lies in the interval $[0, \frac{1}{4} \sum_{j=1}^{4} \log(\frac{1}{4P_j^i})] = [0, \ 0.1757]$, satisfying the condition given in Theorem 1, Lemma 1 guarantees the existence of a positive $u$ such that equality holds for the KL constraint (2). Numerically, it turns out that the solution is $u = 1$ in this case, yielding the probability:

$$Q^i = \frac{1}{\sum_{j=1}^{4} P_j^{\frac{1}{1+1}}} [0.4^{\frac{1}{1+1}}, 0.3^{\frac{1}{1+1}}, 0.2^{\frac{1}{1+1}}, 0.1^{\frac{1}{1+1}}] = [0.3254, 0.2818, 0.2301, 0.1627] \quad (7)$$

It is evident that the resulting probability distribution is more uniform compared to the initial $P^i$. This corresponds to a higher entropy, allowing us to embed more secret bits with $Q^i$ than with $P^i$.

# 4 PRACTICAL CONSIDERATIONS

## 4.1 TOKENIZATION ERROR

LLM-based steganography relies on several assumptions. Firstly, the underlying LLM and the parameters given to both Alice and Bob must be identical. Second, it is essential that Bob's tokenization process matches that intended by Alice. The second assumption is in fact quite subtle, and is complicated by the sub-word tokenizer used in modern pre-trained LLMs. These tokenizers can guarantee that after detokenizing, the original text can be recovered; however, it does not guarantee the tokenizer can always reproduce the same sequence of tokens from the detokenized text. For example, a token sequence Alice generated during the stego-text encoding process is {"This", "mount", "ain", "is", "high"}, resulting in the stego-text containing "This mountain is high", which Bob might incorrectly tokenize to {"This", "mountain", "is", "high"}. In order words, the tokenizer merged "mountain" into a single token rather than the two that the stego-text encoder intended. This issue exists in most of the previous LLM-based steganography approaches (Ziegler et al., 2019; Shen et al., 2020), though it has not been addressed explicitly so far.

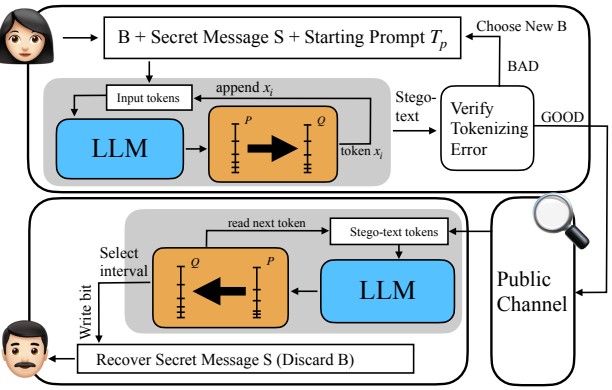

Figure 2: The OD-Stega approach

This tokenization error leads Bob to decode a bit sequence different from the original secret bit sequence. After thorough testing, we found that the likelihood of such errors occurring is proportional to the length of the bit file. In mathematical terms, the relationship can be described as $\epsilon_{tok} = O(n)$, where $n$ is the number of secret bits, and $\epsilon_{tok}$ represents the error rate, measuring the proportion of tests that fail due to tokenization errors relative to the total number of tests.

Since LLMs are computationally demanding, it is not realistic to enumerate all such potential error cases to design strategies to prevent such errors from occurring. Instead, we observe that Alice can in fact verify whether the stego-text can be correctly decoded by Bob since both sides have a copy of the same tokenizer. Based on this observation, we propose the following strategy. We prepend a short sequence of additional $B$ bits to the bit sequence $S$. Alice then iterates among all $B$-bits combinations, and uses $f(T_p, (B, S))$ to produce the stego-text, until she verifies Bob can indeed correctly decode the text. Bob simply discards the beginning $B$ bits after decoding.

Next we determine an appropriate choice for the length of $B$ that guarantees the entire steganography process succeeds with high probability, which we set as $1 - 10^{-8}$ in our work. Our experiments reveal that for LLAMA models, a single bit produces a tokenization error at a rate below $2 \times 10^{-4}$. Since we are essentially making $2^{|B|}$ independent attempts to find a successful embedding, we can ensure that at least one of these attempts does not have any tokenization errors by setting

$$|B| > 3 - \log_2\left(4 - \log_{10} 2 - \log_{10} |S|\right), \tag{8}$$

assuming $B$ is considerably shorter than the length of $S$, which is justified by empirical observation that tokenization errors do not occur very often. The overall OD-Stega approach with this consideration is illustrated in Figure 2.

## 4.2 REDUCE COMPUTATIONAL COMPLEXITY VIA VOCABULARY TRUNCATION

To reduce the computational complexity when the vocabulary set is large, especially when there is a large number of tokens with probabilities near zero, a simple strategy is to truncate the vocabulary in the subsequent processing once a probability distribution has been generated. This strategy has been adopted in Shen et al. (2020). To leverage our optimization formulation, we consider a two-stage

process: first, we truncate the vocabulary, and second, we optimize the probability adjustment on the truncated vocabulary as discussed in the previous section. For this two-stage approach, we establish the KL divergence between the original distribution and the eventual distribution on the truncated vocabulary, given below in Theorem 2.

Let us make the two-stage strategy more precise. We first expand the zero-probability index set $\mathbb{A}_i$ from $[N_i + 1 : N]$ to $[N_\epsilon + 1 : N]$, where $N_\epsilon = \min\{n \mid \sum_{j=1}^{n} P_j^i \geq 1 - \epsilon\}$. This leaves us with a total of $N_\epsilon$ variables. There may not exist an $n$ such that $\sum_{j=1}^{n} P_j^i = 1 - \epsilon$ exactly, but for simplicity, we assume that this can be achieved exactly, meaning that $\sum_{j=1}^{N_\epsilon} P_j^i = 1 - \epsilon$. This assumption is reasonable, since in LLMs, the number of tokens is quite large, and the cutoff value $\epsilon$ is small, therefore, this approximation is usually quite accurate. After the first stage, the variables in the optimization problem are reduced to $[Q_1^i, \cdots, Q_{N_\epsilon}^i]$.

We can now focus on the most likely symbols in the probability list $P_j^i$, $j \in [1 : N_\epsilon]$. We define the re-normalized probability $\hat{P}_j^i(\epsilon) = \frac{1}{1-\epsilon} P_j^i$, which we refer to as an $\epsilon$ cutoff probability of $P^i$. The KL divergences between $P^i$ and its cutoff $\hat{P}^i(\epsilon)$ are

$$D_{KL}(\hat{P}^i(\epsilon)||P^i) = \sum_{j=1}^{N_\epsilon} \hat{P}_j^i(\epsilon) \log\left(\frac{\hat{P}_j^i(\epsilon)}{P_j^i}\right) = \sum_{j=1}^{N_\epsilon} \frac{1}{1-\epsilon} P_j^i \log\left(\frac{\frac{1}{1-\epsilon} P_j^i}{P_j^i}\right) \tag{9}$$

$$= \frac{1}{1-\epsilon} \log\left(\frac{1}{1-\epsilon}\right) \sum_{j=1}^{N_\epsilon} P_j^i = -\log(1-\epsilon) \tag{10}$$

The next theorem establishes the KL divergence between the original distribution $P^i$ and the optimized distribution $Q^i$, the latter of which is obtained by solving the optimization problem in (1)-(5), with $\hat{P}^i(\epsilon)$ replacing $P^i(\epsilon)$.

**Theorem 2** *Let $\hat{P}^i(\epsilon)$ be the $\epsilon$ cutoff probability distribution of $P^i$ and $Q^i$ be the solution of the optimization problem (1)-(5) with the constraint $D_{KL}(Q^i||\hat{P}^i(\epsilon)) \leq \hat{\delta}(\epsilon)$, then it holds that*

$$D_{KL}(Q^i||P^i) = D_{KL}(\hat{P}^i(\epsilon)||P^i) + D_{KL}(Q^i||\hat{P}^i(\epsilon)). \tag{11}$$

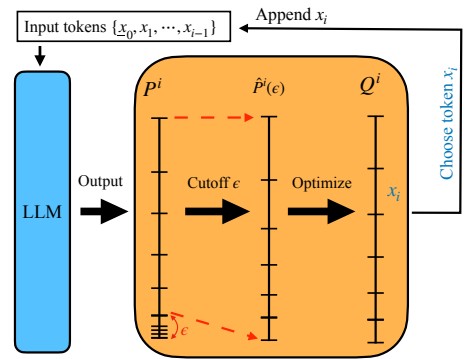

Figure 3: The two-stage design: Vocabulary truncation and distribution optimization

The proof of Theorem 2 can be found in the Appendix. It is well known that the KL divergence is not a true metric since it is not symmetric and does not satisfy the triangular inequality in general. Theorem 2 indicates that, in the specific scenario involving the cutoff probability and optimized counterpart, the KL divergence is in fact additive. Given a total KL budget $\delta$, it is clear that we can determine $\hat{\delta}(\epsilon) = \delta + \log(1-\epsilon)$, where $-\log(1-\epsilon)$ represents the KL divergence between $\hat{P}^i(\epsilon)$ and $P^i$ as given in (10). Since the KL divergence is positive, it is essential to select $\epsilon$ within the range $0 < \epsilon < 1 - e^{-\delta}$ to guarantee that $\hat{\delta}(\epsilon)$ represents a valid KL divergence value.

### 4.3  $\delta$ SELECTION ON THE SEQUENCE LEVEL

Denote the divergence threshold in each time $i$ as $\delta_i$. If $\delta_i$ is set too large, the resulting adjustment to the probability distribution may lead to the selection of unusual tokens, negatively impacting the fluency of the stego-text. This issue is particularly noticeable when dealing with positions that have probability distributions with very low entropy values, i.e., most tokens have near-zero probability and the choices of tokens are almost deterministic. To address this issue, we need to choose $\delta_i$ at the sequence level adaptively to the entropy $H(P^i)$, i.e. $\delta_i = h(H(P^i))$. A simple approach is to set

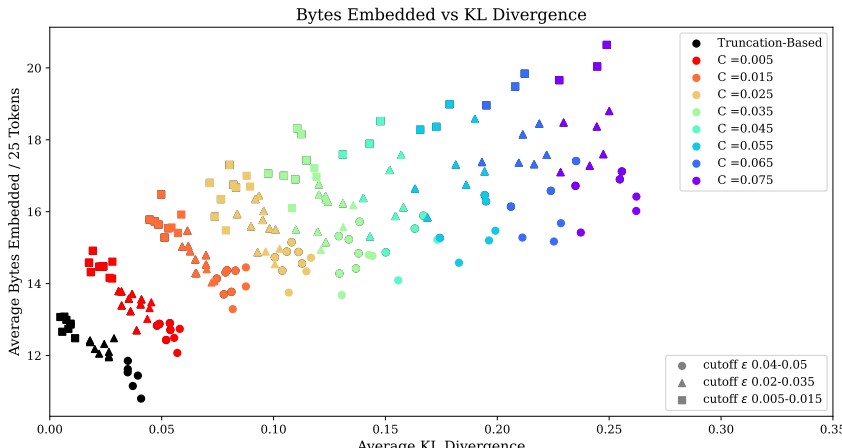

Figure 4: Average (over 100 runs) bytes embedded vs. KL divergence. The colored points represent OD-Stega, while the black data points correspond to the truncation-based method. The parameter $C$ controls the adjustment $\delta_i$ at each time step; as $C$ increases, the distribution diverges further from the natural language distribution.

$\delta_i = C \cdot H(P^i)$ where $C$ is a constant. Furthermore, we introduce another threshold $\alpha$,

$$\delta_i = \begin{cases} C \cdot H(P^i), & \text{if } H(P^i) \geq \alpha \\ 0, & \text{if } H(P^i) < \alpha \end{cases} \tag{12}$$

which means that for the position where $H(P^i)$ falls below this threshold, we set $\delta_i$ to zero.

## 5 EXPERIMENTAL RESULTS

### 5.1 EXPERIMENTAL SETUP

In our experiment, we chose the LLAMA2-7B pretrained model (Touvron et al. (2023)) as our main Large Language Model, employing the SentencePiece tokenizer. This LLM features a vocabulary of 32,000 tokens, facilitating efficient tokenization and diverse text representation.

We performed experiments using a range of starting prompts on different topics of interest. Examples topics include the Olympics, news, technology, and blogs, among others. The prompts usually have 10 to 20 words. Despite their brevity, we demonstrate that OD-Stega can still generate stego-texts that remains relevant to the initial prompt with the assistance of contemporary LLMs.

In our two-stage optimization framework, we typically select a cutoff value $\epsilon$ within the range $(0, 0.05]$, and adjust the constant $C \in [0, 0.2)$ in (12) to control the $\delta_i$ values. Additionally, the threshold $\alpha$ is adjusted within the interval $[0, 2]$ to enhance the optimization procedure. Setting the cutoff $\epsilon$ at its maximum of 0.05 results in the effective elimination of roughly 2000 variables. Moreover, by adjusting the range of $\delta_i$ and $\alpha$ values, we can assess how these values influence both the performance of the generated stego-text and the number of embedded bits.

The primary evaluation metric used in this study is the number of embedded secret bits per token, or equivalent the number of embedded bytes for a fixed number of generated stego-text tokens, for which a higher value indicates more efficient embedding. The quality of the generated stego-text is measured by two metrics: 1) The **KL Divergence** where a lower value implies better imperceptibility; 2) A perception evaluation using **GPT-4** as a human perception surrogate, where we simply ask GPT to determine whether the stego-text is written by human or not. We refer to the approach of Shen et al. (2020) as the truncation-based method, and use it as the main reference to compare with our proposed OD-Stega method.

### 5.2 BITS/TOKEN (EMBEDDING UTILIZATION) VS. KL TRADEOFF

In this experiment, we keep the number of tokens in the stego-text to be 25, but attempt to embed more secret bits than can be embedded with 25 tokens in order to test the limits of the method. By

varying parameter $C$ from 0 to 0.075 and adjusting parameters $\epsilon$ and $\alpha$, we obtained various pairs consisting of number of bytes embedded and the corresponding KL divergence, shown in Figure 4. Different shapes of the data points in this plot correspond to different levels of truncation cut-off value. The highest contour curve predominantly consists of the square points, representing the smallest cutoff category in our experiments, ranging from 0.005 to 0.015. This behavior suggests that, given a fixed KL divergence budget, allocating a larger proportion of the probability distance to the optimization process, rather than to the truncated portion, results in more effective bit embedding.

We observe that as $C$ increases (which corresponds to an increase in $\delta_i$), the data points move linearly towards the upper right, meaning more secret bits are embedded, but the stego-text becomes less natural. On the other hand, the black points representing the truncation-based method (Shen et al., 2020) shifts toward lower right, meaning losses in the embedding capability. From this plot, it is clear that the proposed method has the ability to embed more than 20 bytes while maintaining a KL divergence below 0.25. In comparison to the truncation-based method, at a KL divergence of 0.02, our approach achieves a 1.25-times improvement in bit embedding capacity. At a KL divergence close to 0.06, our method shows an even greater enhancement, achieving a 1.5-times increase in embedding efficiency over the truncated method.

## 5.3 GPT Evaluation

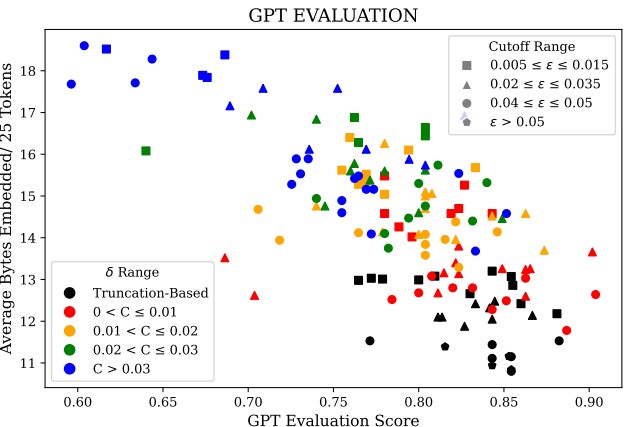

Figure 5: Average (over 100 runs) bytes embedded vs. GPT evaluation score. The colored data points represent the OD method, while the black data points correspond to the truncated-based method with varying cutoff values.

We use the GPT-4 model to evaluate whether our stego-text appears natural and can avoid detection by the human eavesdropper Eve. We instructed GPT to mimic a human evaluator to assess the text and determine if it was likely written by a human, responding with either "yes" or "no". In this experiment, we examined hundreds of generated stego-texts with GPT-4 under various parameters outlined in Section 5.2, with the results displayed in Figure 5. The horizontal axis represents the GPT evaluation score, i.e., the ratio of test cases marked "yes" by GPT in the total number of files evaluated.

Since a higher GPT score indicates a better result, the upper right direction means better performance in this point cloud plot. We first observe that the KL divergence is a relatively accurate measure of human perception. The black data points represent the truncation-based method, which again under-performs. At a GPT evaluation score of approximately 0.77, OD-Stega can attain an embedding rate 1.4 times higher; at around 0.85, OD-Stega even achieves 1.5 times the number of embedded bits than the truncation-based approach. We observe that a cutoff value between 0.01 and 0.3 appears to be suitable for OD-Stega.

Interestingly, at the extreme high GPT score regime, OD-Stega with a small truncation can achieve above 0.9, which the approach with essentially unadjusted LLM distributions (extremely small truncation values and no optimization of the distribution) cannot achieve. In other words, at the extreme regime, the OD-Stega approach can achieve more natural stego-texts than those directly generated from the LLAMA model, viewed from the point of a GPT surrogate.

## 5.4 Examples of Generated Stego-Texts

Figure 6 presents examples of stego-texts generated using our proposed method. Given a secret message $S$ and an initial prompt $T_p$, two text outputs were generated by varying the parameter

values. The prompt, which discusses animals in the Amazon rainforest, yielded distinct results based on the chosen parameter settings. The green text represents coherent and logical content, while the red text deviates from the given topic. The green OD-Stega text, generated using the parameter pair $(C = 0.025, \epsilon = 0.05)$, demonstrates fluency and maintains consistency with the prompt's topic. In contrast, the red OD-Stega text, produced with a larger parameter $C = 0.05$ and higher KL divergence, shows a significant departure from the natural language distribution. Specifically, in the second sentence, the text becomes incoherent, leading to the generation of awkard and off-topic content in this instance.

| Prompt : There are many species of animals living in the Amazon rainforest, including species such as | | |
|---|---|---|
| Secret Message

S :
00111001100010101
01111111001011100
00100011101010101
11100010011010110
01010100111111000
01000000100101110
01101011001100010
101001000.. | $C = 0.025$
$\epsilon = 0.05$ | iguanas, turtles, snakes, birds, and small amphibians. Forest clearing for permanent settlement may create refuge areas for big-footed short-horned grassland ungulates by allowing the trees that otherwise line the field boundaries to be harvested … |
| | $C = 0.05$
$\epsilon = 0.05$ | iguanas, deer, porcupines and armadillos are likely culprits. In 1804, Brother Bernard Hyacinthe Jusserand was appointed Professor of Mathematics. Joseph Girard took over the Dean's role. Walter Goodman replaced Erwin… |

Figure 6: Stego-text examples in different parameters: The green text illustrates more fluent and logically consistent output, while the red text shows incoherent and less natural results.

| Prompt : BREAKING NEWS: Yesterday in Pennsylvania, | | |
|---|---|---|
| Secret Message

S :
0100111011011011
1110111000101111
1010111100000001
1111010010101001
0110101110000111
1000110111000110
1100100101101010
00101011001011.. | $C = 0.05$
$\epsilon = 0.025$ | 13 farmers joined together to fight for the protections that would protect family farm businesses in between fourth quarter calendar quarters rather than incorporate schedule form adjustments available… |
| | Truncation based

$\epsilon = 0.05$ | 10/21, there was a 2-day Exhibition and Conference on Cement and Building Materials Various Densities of Self-Compacting Concrete with Polymeric Modification属于商业化的小麦图… |

Figure 7: Stego-text examples in different methods. The green text illustrates more fluent and logically consistent output, while the red text shows incoherent and less natural results.

Figure 7 presents a comparison between two stego-text generation methods: truncation-based and OD-Stega. The results indicate that when using a larger value of $\epsilon$ in the truncation-based method, the generated text can produce anomalous tokens, as illustrated by the red text in this example. Specifically, with a cutoff of $\epsilon = 0.05$, the truncation-based method starts generating irregular tokens after producing 20 tokens. In contrast, by using a smaller $\epsilon$ and allocating more of the adjustment budget to the optimization stage, as done for the green text, the output appears significantly more natural. More examples can be found in Appendix F.

## 6    CONCLUSION

To embed more secret messages in stego-texts while reducing computational complexity and maintaining near-imperceptibility, we propose the OD-Stega method. This approach optimizes the probability distribution towards a more uniform structure under a perception constraint. Additionally, we address the tokenization errors that often arise in LLM-based steganography due to the use of sub-word tokenizers in modern LLMs. Together with the vocabulary truncation technique, our two-stage embedding process significantly increases the embedding efficiency under the KL divergence constraint, and demonstrates strong imperceptibility performance. We conducted extensive tests and evaluate the outputs both using the KL divergence value and the GPT evaluation. OD-Stega provides a robust solution, enhancing both efficiency and security in LLM-based steganographic embedding.

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

# A    RELATED WORKS

Linguistic Steganography (LS) can be divided into two main areas: modification-based (cover-based) and generation-based (coverless). The modification-based approach conceals secret messages by altering the cover text through synonyms, syntactic changes, and word substitutions (Topkara et al., 2006; Chang & Clark, 2010; Qi et al., 2013; Chang & Clark, 2014). In contrast, the generation-based approach creates stego-texts using methods like Markov chains (Dai et al., 2009; 2010; Moraldo, 2012) and deep learning techniques. With the advancement of generative language models, an increasing number of steganography research efforts now leverage neural networks to produce steganographic texts (Fang et al., 2017; Yang et al., 2018; Ziegler et al., 2019; Xiang et al., 2017; Dai & Cai, 2019; Zhang et al., 2021; Shen et al., 2020; Kaptchuk et al., 2021; Ding et al., 2023; de Witt et al., 2024)

Fang et al. (2017), for instance, explored a block-based methodology in which they designed a text generation model that first partitions the dictionary and allocates a specific code for each word. During the output stage, modified word-level LSTM neural network is utilized to choose words according to the encoded secret information. Their method organizes the vocabulary into subsets, the best word is chosen from a candidate pool based on the encoded bitstream at every generation step. Yang et al. (2018) presented a model that enhances text fluency and security in steganography by encoding each word dynamically based on its conditional probability distribution, employing both fixed-length coding (FLC) and variable-length coding (VLC). Through the use of structures like full binary trees or Huffman trees, this method enhances the naturalness and quality of generated texts while embedding hidden information more effectively.

Ziegler et al. (2019) also utilized GPT-2 to create stego-texts, by proposing a linguistic steganography method that uses arithmetic coding with a pretrained neural language model. This method encodes secret messages by truncating the token distribution to the top $K$ most probable tokens at each generation step, thus minimizing the difference between the conditional probability distributions of steganographic and normal text, achieving close to optimal statistical security. Human evaluations were conducted to confirm that the generated text successfully deceived readers.

Building on Ziegler et al.'s arithmetic coding and truncating probability method, Shen et al. (2020) modified $K$ for each iteration, adjusting the conditional probability threshold with each new token. They claimed to select the smallest $K$ that still ensured near-imperceptibility. Additionally, they employed human evaluations to confirm their findings, demonstrating their method's effectiveness in deceiving eavesdroppers.

Dai & Cai (2019) employed GPT-2 for generating steganographic texts, crafting a novel steganographic mapping to embed secret messages and showcasing that effective mapping increases text security. They also proposed the patient-Huffman algorithm in such setting, which dynamically adjusts the embedding rate through the application of Kullback-Leibler divergence, enhancing both the quality and imperceptibility of steganographic texts. Their approach achieved near-imperceptibility, validated using total variation distance.

Recognizing the informal nature in the treatment of the security aspect of the methods in the studies from natural language processing community Ziegler et al. (2019); Dai & Cai (2019); Shen et al. (2020), the security research community further refined these methods to obtain provably secure protocols (Kaptchuk et al., 2021; Zhang et al., 2021; Ding et al., 2023; de Witt et al., 2024). Zhang et al. (2021) attempted to use grouping to match the granularity of probability to that of the secret message distribution granularity, however, their method is only perfectly secure when the natural language distribution allows such a grouping. Moreover, the grouping operation itelf also leads to a loss of embedding utilization. Kaptchuk et al. (2021) replaced the repeated secret key in Ziegler et al. (2019) with pseudo-random generators, and showed that the resulting protocol is provably secure. However, the arithmetic coding component in Kaptchuk et al. (2021) is a reduced version from the full version, resulting in a slight loss in the embedding utilization. Instead of encrypting the original message and then using the generative model for steganography encoding, Ding et al. (2023) combined the encryption step and the steganography encoding, resulting in another provably secure protocol. The work de Witt et al. (2024) proposed a different approach to couple the message and the stego-text than using arithmetic coding directly.

In this paper, we present our encoding-decoding framework, drawing inspiration from Ziegler et al. (2019) and Shen et al. (2020). We observed that truncating a significant portion of the conditional probability from below leads to a reduction in bits embedded, which improves computational efficiency but reduces capacity. In fact, their approach for embedding long secret messages requires more computation in order to generate long stego-texts. To resolve this issue, we propose a novel method for adjusting the conditional probability to maximize the information embedded while maintaining near imperceptibility. Our results demonstrate that we can embed nearly 1.5 times the amount of bits compared to the previous work.

## B    PROOF OF THEOREM 1

The Lagrangian function of the problem is

$$\mathscr{L} = \sum_{j=1}^{N_i} Q_j^i \log Q_j^i + u \left( \sum_{j=1}^{N_i} Q_j^i \log(\frac{Q_j^i}{P_j^i}) - \delta \right) + \boldsymbol{\lambda}^T(-Q^i) + \omega \left( \sum_{j=1}^{N_i} Q_j^i - 1 \right) \tag{13}$$

where $u, \boldsymbol{\lambda}, \omega$ are the Lagrangian multipliers of constraint (2), (3) and (4), respectively. Then the KKT condition can be derived as follows:

1. Stationarity:

$$\frac{\partial \mathscr{L}}{\partial Q_j^i} = \log Q_j^i + 1 + u \left( \log \frac{Q_j^i}{P_j^i} + 1 \right) - \lambda_j + \omega = 0, \quad \forall j \in [1:N_i] \tag{14}$$

2. Primal feasibility:

$$\begin{cases} \sum_{j=1}^{N_i} Q_j^i \log \frac{Q_j^i}{P_j^i} - \delta \leq 0 \\ Q_j^i \geq 0, \quad \forall j \in [1:N_i] \\ \sum_{j=1}^{N_i} Q_j^i - 1 = 0 \end{cases} \tag{15}$$

3. Dual feasibility:

$$\begin{cases} u \geq 0 \\ \lambda_j \geq 0, \quad \forall j \in [1:N_i] \end{cases} \tag{16}$$

4. Complementary slackness:

$$\begin{cases} u \left( \sum_{j=1}^{N_i} Q_j^i \log \frac{Q_j^i}{P_j^i} - \delta \right) = 0 \\ \lambda_j Q_j^i = 0, \forall j \in [1:N_i] \\ \omega \left( \sum_{j=1}^{N_i} Q_j^i - 1 \right) = 0 \end{cases} \tag{17}$$

Since the optimization problem is convex and clearly feasible, a solution to the KKT condition is also a global optimal solution. We claim the following is a solution to the KKT conditions:

1. Primal variables:
   In case $\delta \in [0, \frac{1}{N_i} \sum_{j=1}^{N_i} \log(\frac{1}{N_i P_j^i})]$, from stationarity in (14),

$$Q_j^i = 2^{\frac{1}{1+u} \left( u \log P_j^i - 1 + \lambda_j - u - \omega \right)} \tag{18}$$

$$= D P_j^{i \frac{u}{1+u}}, \ \forall j \in [1:N_i] \tag{19}$$

where $D = 2^{\frac{-1+\lambda_j - u - \omega}{1+u}}$ is a constant.
Since $\sum_{j=1}^{N_i} Q_j^i = 1$, we can simply rewrite $Q_j^i$ in the form:

$$Q_j^i = \frac{P_j^{i \frac{u}{1+u}}}{\sum_{j=1}^{N} P_j^{i \frac{u}{1+u}}}, \ \forall j \in [1:N_i] \tag{20}$$

In case $\delta > \frac{1}{N_i} \sum_{j=1}^{N_i} \log(\frac{1}{N_i P_j^i})$, we have

$$Q_j^i = \frac{1}{N_i}, \ \forall j \in [1 : N_i] \tag{21}$$

2. Dual variables:

$$\begin{cases} u \begin{cases} \geq 0, \ \delta \in [0, \frac{1}{N_i} \sum_{j=1}^{N_i} \log(\frac{1}{N_i P_j^i})] \\ = 0, \ \delta > \frac{1}{N_i} \sum_{j=1}^{N_i} \log(\frac{1}{N_i P_j^i}) \end{cases} \\ \lambda_j = 0, \ \forall j \in [1 : N_i] \\ \omega = \frac{1}{1+u} \left( -1 + \log(\sum_{j=1}^{N_i} P_j^{i\frac{u}{1+u}}) \right) \end{cases} \tag{22}$$

It is straightforward to verify all the KKT conditions are satisfied, except the dual feasibility condition $u \geq 0$, which we prove in Lemma 1 next.

## C  PROOF OF LEMMA 1

First, we note that $\lim_{u \to 0} D_{KL}(Q^i || P^i) = \frac{1}{N_i} \sum_{j=1}^{N_i} \log(\frac{1}{N_i P_j^i})$ and $\lim_{u \to \infty} D_{KL}(Q^i || P^i) = 0$, because

$$\lim_{u \to 0} Q_j^i = \frac{1}{N_i} \Rightarrow \lim_{u \to 0} D_{KL}(Q^i || P^i) = \sum_{j=1}^{N_i} \frac{1}{N_i} \log \left( \frac{\frac{1}{N_i}}{P_j^i} \right) = \frac{1}{N_i} \sum_{j=1}^{N_i} \log \left( \frac{1}{N_i P_j^i} \right) \tag{23}$$

$$\lim_{u \to \infty} Q_j^i = P_j^i \Rightarrow \lim_{u \to \infty} D_{KL}(Q^i || P^i) = \sum_{j=1}^{N_i} P_j^i \log \left( \frac{P_j^i}{P_j^i} \right) = 0. \tag{24}$$

Second, note that that $D_{KL}(Q^i || P^i)$ is continuous in $u \geq 0$. To see this, consider $P_j^i$ as the known distribution value, $Q_j^i$ is continuous in $u \geq 0$ because $\frac{u}{1+u}$ is continuous in $\mathbb{R} \setminus \{-1\}$. In addition, $Q_j^i$ will not be zero for all $j \in [1 : N_i]$, which indicates that $\log(\frac{Q_j^i}{P_j^i})$ is continuous. Therefore, $D_{KL}(Q^i || P^i)$ is also continuous in $u \geq 0$ since the function is a linear combination of continuous functions.

Lemma 2, which is proved below, states that $D_{KL}(Q^i || P^i)$ is non-increasing in $u$ for $u \geq 0$. By the Intermediate-Value Theorem (IVT), it is clear that there exists a positive $u$ such that the KL divergence is equal to the given $\delta \in [0, \frac{1}{N_i} \sum_{j=1}^{N_i} \log(\frac{1}{N_i P_j^i})]$.

## D  PROOF OF LEMMA 2

Here we show that $D_{KL}(Q^i || P^i)$ is non-increasing in $u$ for $u \geq 0$, by analyzing the derivative as follows:

$$\frac{\partial D_{KL}(Q^i || P^i)}{\partial u} = \frac{\partial}{\partial u} \left( \sum_{j=1}^{N_i} Q_j^i \log \frac{Q_j^i}{P_j^i} \right) = \sum_{j=1}^{N_i} \frac{\partial}{\partial u} \left( Q_j^i \log \frac{Q_j^i}{P_j^i} \right) \tag{25}$$

$$= \sum_{j=1}^{N_i} \left\{ \frac{\partial}{\partial u} \left( \frac{P_j^{i\frac{u}{1+u}}}{\sum_{k=1}^{N_i} P_k^{i\frac{u}{1+u}}} \right) \log \left( \frac{P_j^{i\frac{-1}{1+u}}}{\sum_{k=1}^{N_i} P_k^{i\frac{u}{1+u}}} \right) + \right.$$

$$\left. \left( \frac{P_j^{i\frac{u}{1+u}}}{\sum_{k=1}^{N_i} P_k^{i\frac{u}{1+u}}} \right) \frac{\partial}{\partial u} \log \left( \frac{P_k^{i\frac{-1}{1+u}}}{\sum_{k=1}^{N_i} P_k^{i\frac{u}{1+u}}} \right) \right\} \tag{26}$$

$$= \sum_{j=1}^{N_i} \left\{ \left[ \left( \sum_{k=1}^{N_i} P_j^{i\frac{u}{1+u}} \right)^{-2} \left( \frac{1}{1+u} \right)^2 P_j^{i\frac{u}{1+u}} \left( \sum_{k=1}^{N_i} P_j^{i\frac{u}{1+u}} \log(\frac{P_j^i}{P_k^i}) \right) \right] \log \left( \frac{P_j^{i\frac{-1}{1+u}}}{\sum_{k=1}^{N_i} P_k^{i\frac{u}{1+u}}} \right) \right.$$

$$
+ \left( \frac{P_j^{i\,\frac{u}{1+u}}}{\sum_{k=1}^{N_i} P_k^{i\,\frac{u}{1+u}}} \right)
$$

$$
\cdot \left[ \left( \frac{\sum_{k=1}^{N_i} P_k^{i\,\frac{u}{1+u}}}{P_j^{i\,\frac{-1}{1+u}}} \right) \left( \sum_{k=1}^{N_i} P_j^{i\,\frac{u}{1+u}} \right)^{-2} \left( \frac{1}{1+u} \right)^2 P_j^{i\,\frac{-1}{1+u}} \left( \sum_{k=1}^{N_i} P_k^{i\,\frac{u}{1+u}} \log(\frac{P_j^i}{P_k^i}) \right) \right] \right\} \tag{27}
$$

$$
= \sum_{j=1}^{N_i} \left( \sum_{k=1}^{N_i} P_k^{i\,\frac{u}{1+u}} \log(\frac{P_j^i}{P_k^i}) \right) \left\{ T^{-2} \left( \frac{1}{1+u} \right)^2 P_j^{i\,\frac{u}{1+u}} \log \left( \frac{P_j^{i\,\frac{-1}{1+u}}}{T} \right) + \right.
$$

$$
\left. \left( \frac{P_j^{i\,\frac{u}{1+u}}}{T} \right) \left( \frac{T}{P_j^{i\,\frac{-1}{1+u}}} \right) T^{-2} \left( \frac{1}{1+u} \right)^2 P_j^{i\,\frac{-1}{1+u}} \right\} \tag{28}
$$

$$
= T^{-2} \left( \frac{1}{1+u} \right)^2 \sum_{j=1}^{N_i} \left\{ P_j^{i\,\frac{u}{1+u}} \left( \sum_{k=1}^{N_i} P_k^{i\,\frac{u}{1+u}} \log(\frac{P_j^i}{P_k^i}) \right) \left( \log \left( \frac{P_j^{i\,\frac{-1}{1+u}}}{T} \right) + 1 \right) \right\} \tag{29}
$$

$$
= T^{-2} \left( \frac{1}{1+u} \right)^2 \sum_{j=1}^{N_i} \left\{ \left( \sum_{k=1}^{N_i} (P_j^i P_k^i)^{\frac{u}{1+u}} \log(\frac{P_j^i}{P_k^i}) \right) \log \left( \frac{2 P_j^{i\,\frac{-1}{1+u}}}{T} \right) \right\} \tag{30}
$$

$$
= T^{-2} \left( \frac{1}{1+u} \right)^2 \sum_{j=1}^{N_i} \sum_{k=1}^{N_i} B_{jk} \log \left( \frac{2 P_j^{i\,\frac{-1}{1+u}}}{T} \right) \tag{31}
$$

$$
= T^{-2} \left( \frac{1}{1+u} \right)^2 \sum_{\substack{j,k=1 \\ j \neq k \\ P_j^i \geq P_k^i}}^{N_i} B_{jk} \left( \log \left( \frac{2 P_j^{i\,\frac{-1}{1+u}}}{T} \right) - \log \left( \frac{2 P_k^{i\,\frac{-1}{1+u}}}{T} \right) \right) \tag{32}
$$

$$
= T^{-2} \left( \frac{1}{1+u} \right)^2 \sum_{\substack{j,k=1 \\ j \neq k \\ P_j^i \geq P_k^i}}^{N_i} B_{jk} \left( \frac{-1}{1+u} \right) \log \left( \frac{P_j^i}{P_k^i} \right) \leq 0 \tag{33}
$$

where $T = \sum_{k=1}^{N_i} P_k^{i\,\frac{u}{1+u}}$ and $B_{jk} = (P_j^i P_k^i)^{\frac{u}{1+u}} \log \left( \frac{P_j^i}{P_k^i} \right)$. Eq. (32) follows from $B_{jk} = -B_{kj}, \forall j \neq k$ and $B_{jk} = 0, \forall j = k$. The only negative term in (33) is $\frac{-1}{1+u}$, since $B_{jk}$ and $\log(\frac{P_j^i}{P_k^i})$ are both positive in the case $P_j^i \geq P_k^i$. This proves the inequality. It follows that that $D_{KL}(Q^i || P^i)$ is non-increasing in $u$ for $u \geq 0$.

## E  PROOF OF THEOREM 2

The result in Theorem 1 shows that the solution to the optimization problem with constraint $D_{KL}(Q^i || \hat{P}^i(\epsilon)) \leq \hat{\delta}(\epsilon)$ is:

$$
Q_j^i = \begin{cases} \frac{\hat{P}_j^i(\epsilon)^{\frac{\hat{u}(\epsilon)}{1+\hat{u}(\epsilon)}}}{\sum_{j=1}^{N_\epsilon} \hat{P}_j^i(\epsilon)^{\frac{\hat{u}(\epsilon)}{1+\hat{u}(\epsilon)}}} & , \forall \, \hat{\delta}(\epsilon) \in [0, \frac{1}{N_\epsilon} \sum_{j=1}^{N_\epsilon} \log(\frac{1}{N_\epsilon \hat{P}_j^i})] \\ \frac{1}{N_\epsilon} & , \text{otherwise} \end{cases} \tag{34}
$$

$$
= \begin{cases} \frac{P_j^{i\,\frac{\hat{u}(\epsilon)}{1+\hat{u}(\epsilon)}}}{\sum_{j=1}^{N_\epsilon} P_j^{i\,\frac{\hat{u}(\epsilon)}{1+\hat{u}(\epsilon)}}} & , \forall \, \hat{\delta}(\epsilon) \in [0, \frac{1}{N_\epsilon} \sum_{j=1}^{N_\epsilon} \log(\frac{1-\epsilon}{N_\epsilon P_j^i})] \\ \frac{1}{N_\epsilon} & , \text{otherwise} \end{cases} \tag{35}
$$

for all $j \in [1 : N_\epsilon]$ and for some positive $\hat{u}(\epsilon)$. In addition, Lemma 1 states that when $\hat{\delta}(\epsilon) \in [0, \frac{1}{N_\epsilon} \sum_{j=1}^{N_\epsilon} \log(\frac{1}{N_\epsilon \hat{P}_j^i})]$, the obtained solution ensures that the KL divergence $D_{KL}(Q^i || \hat{P}^i(\epsilon))$ is

equal to the given constraint $\hat{\delta}(\epsilon)$. In the following, we show that in this case, the KL divergence is additive, which means that the divergence between $Q^i$ and $P^i$ is the sum of the divergence between $Q^i$ and $\hat{P}^i(\epsilon)$ and between $\hat{P}^i(\epsilon)$ and $P^i$.

$$\hat{\delta}(\epsilon) = \sum_{j=1}^{N_\epsilon} P_j^i \log\left(\frac{P_j^i}{\hat{P}_j^i(\epsilon)}\right) \tag{36}$$

$$= \sum_{j=1}^{N_\epsilon} \left(\frac{\hat{P}_j^i(\epsilon)^{\frac{\hat{u}(\epsilon)}{1+\hat{u}(\epsilon)}}}{\sum_{j=1}^{N_\epsilon} \hat{P}_j^i(\epsilon)^{\frac{\hat{u}(\epsilon)}{1+\hat{u}(\epsilon)}}}\right) \log\left(\frac{\hat{P}_j^i(\epsilon)^{\frac{\hat{u}(\epsilon)}{1+\hat{u}(\epsilon)}}}{\hat{P}_j^i(\epsilon) \sum_{j=1}^{N_\epsilon} \hat{P}_j^i(\epsilon)^{\frac{\hat{u}(\epsilon)}{1+\hat{u}(\epsilon)}}}\right) \tag{37}$$

$$= \sum_{j=1}^{N_\epsilon} \left(\frac{P_j^{i\frac{\hat{u}(\epsilon)}{1+\hat{u}(\epsilon)}}}{\sum_{j=1}^{N_\epsilon} P_j^{i\frac{\hat{u}(\epsilon)}{1+\hat{u}(\epsilon)}}}\right) \log\left(\frac{P_j^{i\frac{\hat{u}(\epsilon)}{1+\hat{u}(\epsilon)}}}{\frac{1}{1-\epsilon} P_j^i \sum_{j=1}^{N_\epsilon} P_j^{i\frac{\hat{u}(\epsilon)}{1+\hat{u}(\epsilon)}}}\right) \tag{38}$$

$$= \sum_{j=1}^{N_\epsilon} \left(\frac{P_j^{i\frac{\hat{u}(\epsilon)}{1+\hat{u}(\epsilon)}}}{\hat{T}(\epsilon)}\right) \log\left(\frac{(1-\epsilon) P_j^{i\frac{-1}{1+\hat{u}(\epsilon)}}}{\hat{T}(\epsilon)}\right) \tag{39}$$

$$= \sum_{j=1}^{N_\epsilon} \left(\frac{P_j^{i\frac{\hat{u}(\epsilon)}{1+\hat{u}(\epsilon)}}}{\hat{T}(\epsilon)}\right) \left(\log(1-\epsilon) + \log\left(\frac{P_j^{i\frac{-1}{1+\hat{u}(\epsilon)}}}{\hat{T}(\epsilon)}\right)\right) \tag{40}$$

$$= \sum_{j=1}^{N_\epsilon} \left(\frac{P_j^{i\frac{\hat{u}(\epsilon)}{1+\hat{u}(\epsilon)}}}{\hat{T}(\epsilon)}\right) \log\left(\frac{P_j^{i\frac{-1}{1+\hat{u}(\epsilon)}}}{\hat{T}(\epsilon)}\right) + \frac{\log(1-\epsilon)}{\hat{T}(\epsilon)} \sum_{j=1}^{N_\epsilon} P_j^{i\frac{\hat{u}(\epsilon)}{1+\hat{u}(\epsilon)}} \tag{41}$$

$$= D_{KL}(Q^i||P^i) + \log(1-\epsilon) \tag{42}$$

$$= D_{KL}(Q^i||P^i) - D_{KL}(\hat{P}^i(\epsilon)||P^i) \tag{43}$$

$$\Rightarrow D_{KL}(Q^i||P^i) = D_{KL}(\hat{P}^i(\epsilon)||P^i) + D_{KL}(Q^i||\hat{P}^i(\epsilon)) \tag{44}$$

where $\hat{T}(\epsilon) = \sum_{j=1}^{N_\epsilon} P_j^{i\frac{\hat{u}(\epsilon)}{1+\hat{u}(\epsilon)}}$ in (39).

# F    MORE EXAMPLES

The example presented in Figure 8 illustrates the generated stego-text by fixing the parameter $C = 0.025$ and comparing the results between an extremely small cutoff value and a typical cutoff value of $\epsilon = 0.05$. In this instance, the green text, corresponding to the normal cutoff, appears logical and coherent, whereas the red text exhibits uncommon word choices after the generation of 10 tokens. This example strengthens our conclusion in Section 5.3, where the cloud plot illustrated that extremely low cutoff values resulted in lower GPT evaluation scores. This occurs because such tokens are not truncated, and the likelihood of being chosen increases after optimization.

In Figure 9 we show more examples of generated stego-texts using the proposed OD-Stega approach with various parameters. It can be seen that as the $(C, \epsilon)$ parameters increase, the embedding capability increases. The generated stego-texts mostly remain fluent in this parameter range.

| Prompt : In the recent Tokyo 2024 Olympics, the most notable event was |
|---|

| Secret Message | | |
|---|---|---|
| S : 000110111001110110 010000110101100111 111101100011001010 001000110111111011 010011001010101100 111011100001100100 001011001111011011 000101000111100101 100111010111001011 1100101111… | $C = 0.025$ $\epsilon = 0.05$ | the 100-meter run won by Grenada's Everard Spicer with a time of 20.61 to narrowly beat first-place qualifier Нагіша турнина … |
| | $C = 0.025$ $\epsilon = 0.005$ | the 10-meter shooting range part of… Page 3, line 21, [Schedule 1], leave out paragraph (c) and add ""to whose ill-health"", after "maintenance-mental". Remove out-of-date references to the North Shore City Council led … |

Figure 8: Stego-text examples in different cutoffs. The green text illustrates more fluent and logically consistent output, while the red text shows incoherent and less natural results.

| Bytes Embedded | Parameters $(C, \epsilon)$ | Prompt + Stego-text |
|---|---|---|
| 11 Bytes | (0.005, 0.005) | In the recent Tokyo 2024 Olympics, the most notable event was the 100 meter men's final between two former WORLD youth medallists. There was a semi-final … |
| 13 Bytes | (0.005, 0.005) | There are many species of animals living in the Amazon rainforest, including species such as iguanas and tree puff-legs, 11 of which are classified as Critically Endangered… |
| 9 Bytes | (0.005, 0.025) | In this blog post, I would like to recount an event that happened to me the other day. I was leaving my house when 14 year-old Hannah spotted me and said: 'Hey, have you got a minute?' … |
| 11 Bytes | (0.005, 0.030) | I went to this restraunt the other day, and I would rate its food 10 out of 10. The meals are fantastic and the response in service is awesome. Staff and students… |
| 11 Bytes | (0.005, 0.040) | Over the next few days, the weather will be 1 to 5 degrees C above average for the northern hemisphere over its 20-year period. But could… |
| 11 Bytes | (0.005, 0.040) | Over the next few days, the weather will be icy. Daytime temperature will reach 35 °c and the rail gauge will remain the same as existing lines so international … |
| 12 Bytes | (0.005, 0.045) | In this blog post, I would like to recount an event that happened to me the other day. I was leaving my house when 150 feet away an old man carrying a wheel barrow. He turned and stopped in front of me and exclaimed … |
| 11 Bytes | (0.005, 0.045) | In this blog post, I would like to recount an event that happened to me the other day. I was leaving my house when 2 CMPD motorcycle officers came out and asked if I had any alcohol on me and then told me that they … |
| 10 Bytes | (0.015, 0.045) | In the recent Tokyo 2024 Olympics, the most notable event was the 100 metres final. The United States did not send track star Tommie Smith and his silver medal to represent their… |
| 13 Bytes | (0.015, 0.010) | BREAKING NEWS: Yesterday in Pennsylvania, 190 Colorado immigrant detainees had been released pending federal reviews of their cases but many had returned to… |
| 10 Bytes | (0.025, 0.045) | There are many species of animals living in the Amazon rainforest, including species such as iguanas, arapas and manakins. The landscape also includes wonderful beaches like Manzanillo, Punta … |
| 12 Bytes | (0.025, 0.010) | BREAKING NEWS: Yesterday in Pennsylvania, 120 residents were evacuated after fuel started leaking from the site. "Those are crimes and … |
| 14 Bytes | (0.035, 0.045) | Due to recent advances in technology, 2.9 million African households now enjoy access to electricity after the Millennium. One day in future… |
| 15 Bytes | (0.035, 0.010) | Over the next few days, the weather will be icy again... Crazy it's so hot here today, it doesn't seem reasonable to just spend a day at … |

Figure 9: Stego-text examples in different pair of parameters $(C, \epsilon)$ and length of secret message embedded.

