# OpenReview forum: "OD-Stega: LLM-Based Near-Imperceptible Steganography via Optimized Distributions"
_ICLR.cc/2025/Conference — Submitted to ICLR 2025_

### Official Review · Reviewer_PEMr · 2024-10-27

**Soundness:** 2
**Presentation:** 3
**Contribution:** 2
**Rating:** 5
**Confidence:** 3

**Summary:**

The authors propose a new form of LLM-based steganography algorithm, with the added goal that the size of the stego-text should be as small as possible. To achieve that, the authors modify the token probability distribution to increase the entropy of the possible choices, therefore increasing the amount of message that can be secretly embedded. However, to maintain security, this must be done while keeping the new distribution relatively similar to the original one to avoid distinguishing attacks. These two contradicting objectives are modeled into a convex optimization problem.

The authors provide a small study of so-called tokenization error issue (present already in state-of-the-art). In addition, to avoid computational complexity to explode in some cases, they leverage a simple and existing truncation strategy of the vocabulary. Finally, they adaptatively control the divergence threshold to avoid special situations where very unusual tokens might be selected.

Experiments have been conducted on LLAMA2-7B with various starting prompts. Examples of stego-test with different parameters values are exhibited.

**Strengths:**

- the general idea to change the token probability distribution is interesting and it is reasonable to think that it should lead to more compact stego-text.

- the paper improves over the truncation-based method

**Weaknesses:**

- I am not really convinced by the evaluation metric for the security part. You check 1) KL divergence and 2) if GPT4 believes the stego-text was written by a human. The first one is quantitative and we can't relate its value to actual security in practice. The second one is very subjective and might completely miss obvious distinguishing patterns. I understand that maximizing these metrics goes towards higher security, but one has no idea if the final generated stego-text could be detected with advanced statistical attacks. Maybe take as example constructions with some provably security https://eprint.iacr.org/2021/686.pdf (CCS 2021)

- Maybe some argumentation is lacking to explain in what practical cases is LLM-based steganography needed ?

**Questions:**

-  looking at figure 4, it would have been great to represent which of these points actually lead to incoherent texts (as shown in figure 6 and 7)

- It would make sense to keep black color for "truncation method" in all your figures, it would be clearer for the reader.

---

> ### Author Response · Authors · 2024-11-20
> **Summary response**
>
> We first thank the reviewer for recognizing the strength of our work. Before proceeding to the item-by-item response, we clarify a few points in a summary below, which were not explained in detail due to space constraints.
>
> 1. Our proposed approach includes the perfectly secure case as a special case. Particularly, note that we assume the secret message has been encrypted. Therefore, the secret message are iid binary (information-theoretic or computational, depending on the encryption process). If we choose the parameter $\delta$ to be zero, i.e., not to increase the embedding utilization by optimizing the distribution, then it is a perfectly secure protocol (almost identical to Meteor).
>
> 2. Our proposed approach essentially provides additional flexibility for system designers to choose a tradeoff point between security and embedding utilization. If in an application, the eavesdropper is resource-limited (or a person paying little attention), then clearly the designer can choose a larger value of $\delta$ to allow more KL-divergence; on the other hand, in other applications, if the security requirement is very high, then $\delta$ should be chosen to be zero, making it completely secure. This additional flexibility is the most important contribution of our work.
>
> 3. The proposed approach can also be easily combined with other techniques, e.g., Discop, Meteor, and ADG. Note that these methods rely on some additional secret keys/randomness, so the overall architecture is somewhat different, however, they all utilize the Ziegler el al. key idea together with some clever randomization techniques. We can optimize the distributions and use the optimized one in Discop, Meteor, or ADG to increase the embedding utilization. Our approach should not be viewed as directly competing with those new er methods but can be viewed as a method to allow further flexibility in them. In other words, our design offers a broader range of design points between a perfect secure solution and the maximum compact embedding solution, on most "base" methods.
>
> 4. Near-perfectly recure steganography is well-accepted in the literature. It had been used widely before coverless steganography became popular. With a covertext, small changes to the covertext always introduce small distribution changes, and near-perfect security has been widely accepted. With more recent coverless steganography, it becomes possible to have "perfect security", such as Discop and Meteor. However, for the coverless setting, "near-imperceptable" steganography was also introduced by Dai and Cai in ACL-19, and it is well-motivated and well-accepted as humans cannot distinguish "perfect security" vs. "near-perfect security".
>
> 5. Some more recent works exist in the literature, however, the improvements over Ziegler's method are minor. For example, both Discop and Meteor are in fact simple variations of Ziegler et al.'s breakthrough idea of using arithmetic coding for steganography. As commented in the Meteor paper, Ziegler's approach is not perfectly secure only due to the reuse of randomness. A minor change to Ziegler's approach will make it perfectly secure: use a random secret key to encrypt the cleartext message (e.g., XOR), then the encrypted secret message can be stega-coded using Ziegler's arithmetic coding approach; the secret key needs to be shared with Bob, but Eve does not have it. Meteor essentially uses this approach, with a simplified arithmetic code that introduces some additional inefficiency (see comments in the Discop paper). Discop uses the random key to instead randomize the Huffman tree (approximately equivalent to fixed-decision arithmetic coding). These newer approaches therefore do not fundamentally impact the performance of Ziegler's, and in fact sometimes they suffer from under-utilization. Ziegler el al.'s work and Dai and Cai's works give the most fundamental methodology, and comparison with them allows a clear picture as to where the strength of our work is, without the distracting factors of randomization techniques in the mix. We can add more comments on the relation if the paper can be accepted.
>
> 6. Even "perfectly secure" protocols such as Discop and Meteor are in fact still not perfectly secure in the strict sense. Firstly, the use of a pseudo-random generator implies that they are only computationally secure, but not information-theoretically secure. Secondly, it is impossible to completely remove the bias in the language model itself, and the "provably secure" protocols assumed language models have no distribution bias, but in practice, this is likely not so. Thirdly, the practical implementation (either Arithmetic coding or the Huffman tree) implies that some small distribution shift will be induced. Therefore, perfect security is only theoretically proved, but not in practice. If we can accept this small security leakage, we do not need to focus singularly on the perfect security to start with, as long as we can control it well.

---

> > ### Author Response · Authors · 2024-11-20
> >
> > 1. Regarding the evaluation metric: Please refer to the summary above first, which explains some of the concerns raised by the reviewer. We would like to emphasize that the main contribution is that the proposed approach in fact allows flexible choice in choosing the tradeoff between security and utilization. By turning the knot $\delta$, we can choose a completely secure protocol with lower utilization, or a less secure one with higher utilization. We observe that in our evaluation, under certain threshold, there is very little difference in both KL and the GPT-based evaluation. However, this knob $\delta$ is really a design parameter, that the application designer should choose for their particular use scenario. If in an application, the eavesdropper is resource-limited, then clearly the designer can choose a larger value of $\delta$; on the other hand, in another application, if the security requirement is very high, then $\delta$ should be chosen to be zero, making it completely secure. We feel this flexibility is the most important contribution. We can attempt to argue that the method is very secure with a very small $\delta$, but that will distract from our main contribution. We also should note that there are recent works that claim perfect security, but in practice, they lead to some security leakage. Please refer to Discop paper for some comments on Meteor and ADG.
> >
> > 2. Regarding the practical cases is LLM-based steganography needed:  We can certainly add an explanation on this, but we feel prior works by Ziegler, Dai and Cai, have already mostly explained why using language models for steganography is beneficial. Essentially, this approach has been shown to be most efficient in terms of security and utilization. Moreover, a smaller language model tends to introduce model bias (i.e., generating unrealistic text by itself), but large language models have mostly resolved this bias issue, therefore, we would need to rely on large models.
> >
> > 3. Regarding Fig. 4: Each point represented the average of 100 different tests, so we couldn’t find a good way to show which ones led to incoherent texts. We have, however, received a few comments asking to demonstrate the frequency of these incoherent texts in a better way, so we will try to resolve this in the revision if it is accepted.
> >
> > 4. Regarding the inconsistency in the figure: Thanks for noting this. It was an oversight and it has been corrected.

---

### Official Review · Reviewer_giCB · 2024-10-30

**Soundness:** 3
**Presentation:** 4
**Contribution:** 2
**Rating:** 3
**Confidence:** 4

**Summary:**

The paper proposes an LLM-based text steganography method OD-Stega, using an improved adaptive arithmetic coding decoder. The authors considered several practical issues such as tokenization error and computational complexity, and they provided strategies to improve efficiency and reliability.

**Strengths:**

1. The paper is well-written. The combination of lemma and examples makes the paper easy to read.
2. Consideration of decoding errors is encouraged.
3. A two-dimensional imperceptibility metric for stegotexts was realized using KL divergence and GPT-4 evaluation.

**Weaknesses:**

1. The steganographic methods cited and compared in this paper only cover approaches developed before 2020. In recent years, many researchers have proposed perfectly secure steganographic methods based on large language models (LLMs) [1][2][3][4][5]. These methods achieve distribution preservation and make efficient use of the information entropy in the prediction distribution of LLMs, which means that these methods modify the KL divergence to almost 0, while utilization of entropy is close to 1. In my view, the proposed OD-Stega method does not offer any advantage in terms of security, and its embedding rate and runtime efficiency have not been compared with these recent methods. The novelty of this paper is therefore questionable. I suggest the authors add a comparison with these methods, focusing on the trade-off between the proposed method's modification of KL divergence and embedding capacity.

- [1] Zhang, Siyu, et al. "Provably Secure Generative Linguistic Steganography." Findings of the Association for Computational Linguistics: ACL-IJCNLP 2021. 2021.
- [2] Kaptchuk, Gabriel, et al. "Meteor: Cryptographically secure steganography for realistic distributions." Proceedings of the 2021 ACM SIGSAC Conference on Computer and Communications Security. 2021.
- [3] de Witt, Christian Schroeder, et al. "Perfectly secure steganography using minimum entropy coupling." arXiv preprint arXiv:2210.14889 (2022).
- [4] Ding, Jinyang, et al. "Discop: Provably secure steganography in practice based on" distribution copies"." 2023 IEEE Symposium on Security and Privacy (SP). IEEE, 2023.
- [5] Zhang, Xin, et al. "Provably secure public-key steganography based on elliptic curve cryptography." IEEE Transactions on Information Forensics and Security (2024).

2. Lack of steganalysis experiments. The paper uses two metrics, KLD and GPT Evaluation Score, to detect the imperceptibility of stegotext, but it lacks the most direct experiments using steganalysis tools as classifiers between covertexts and stegotexts. Existing linguistic steganalysis tools like [6][7] can only achieve about 50% accuracy when detecting the above provably secure methods [1][2][3][4][5], which is similar to random guessing. Whether the method proposed in the paper can achieve similar safety performance is an important metric.

- [6] Yang, Zhongliang, Yongfeng Huang, and Yu-Jin Zhang. "A fast and efficient text steganalysis method." IEEE Signal Processing Letters 26.4 (2019): 627-631.
- [7] Niu, Yan, et al. "A hybrid R-BILSTM-C neural network based text steganalysis." IEEE Signal Processing Letters 26.12 (2019): 1907-1911.

**Questions:**

1. The paper refers to the proposed LLM-sampling-based linguistic steganography as coverless steganography. Why is the text generated by normal sampling of LLM not considered by the authors to be cover text?
2. Does the receiver need to know the length of B during decoding? On average, how many repetitions are required for each generation by the sender?

---

> ### Author Response · Authors · 2024-11-20
> **Summary response**
>
> We first thank the reviewer for recognizing the strength of our work. Before proceeding to the item-by-item response, we clarify a few points in a summary below, which were not explained in detail due to space constraints.
>
> 1. Our proposed approach includes the perfectly secure case as a special case. Particularly, note that we assume the secret message has been encrypted. Therefore, the secret message are iid binary (information-theoretic or computational, depending on the encryption process). If we choose the parameter $\delta$ to be zero, i.e., not to increase the embedding utilization by optimizing the distribution, then it is a perfectly secure protocol (almost identical to Meteor).
>
> 2. Our proposed approach essentially provides additional flexibility for system designers to choose a tradeoff point between security and embedding utilization. If in an application, the eavesdropper is resource-limited (or a person paying little attention), then clearly the designer can choose a larger value of $\delta$ to allow more KL-divergence; on the other hand, in other applications, if the security requirement is very high, then $\delta$ should be chosen to be zero, making it completely secure. This additional flexibility is the most important contribution of our work.
>
> 3. The proposed approach can also be easily combined with other techniques, e.g., Discop, Meteor, and ADG. Note that these methods rely on some additional secret keys/randomness, so the overall architecture is somewhat different, however, they all utilize the Ziegler el al. key idea together with some clever randomization techniques. We can optimize the distributions and use the optimized one in Discop, Meteor, or ADG to increase the embedding utilization. Our approach should not be viewed as directly competing with those new er methods but can be viewed as a method to allow further flexibility in them. In other words, our design offers a broader range of design points between a perfect secure solution and the maximum compact embedding solution, on most "base" methods.
>
> 4. Near-perfectly recure steganography is well-accepted in the literature. It had been used widely before coverless steganography became popular. With a covertext, small changes to the covertext always introduce small distribution changes, and near-perfect security has been widely accepted. With more recent coverless steganography, it becomes possible to have "perfect security", such as Discop and Meteor. However, for the coverless setting, "near-imperceptable" steganography was also introduced by Dai and Cai in ACL-19, and it is well-motivated and well-accepted as humans cannot distinguish "perfect security" vs. "near-perfect security".
>
> 5. Some more recent works exist in the literature, however, the improvements over Ziegler's method are minor. For example, both Discop and Meteor are in fact simple variations of Ziegler et al.'s breakthrough idea of using arithmetic coding for steganography. As commented in the Meteor paper, Ziegler's approach is not perfectly secure only due to the reuse of randomness. A minor change to Ziegler's approach will make it perfectly secure: use a random secret key to encrypt the cleartext message (e.g., XOR), then the encrypted secret message can be stega-coded using Ziegler's arithmetic coding approach; the secret key needs to be shared with Bob, but Eve does not have it. Meteor essentially uses this approach, with a simplified arithmetic code that introduces some additional inefficiency (see comments in the Discop paper). Discop uses the random key to instead randomize the Huffman tree (approximately equivalent to fixed-decision arithmetic coding). These newer approaches therefore do not fundamentally impact the performance of Ziegler's, and in fact sometimes they suffer from under-utilization. Ziegler el al.'s work and Dai and Cai's works give the most fundamental methodology, and comparison with them allows a clear picture as to where the strength of our work is, without the distracting factors of randomization techniques in the mix. We can add more comments on the relation if the paper can be accepted.
>
> 6. Even "perfectly secure" protocols such as Discop and Meteor are in fact still not perfectly secure in the strict sense. Firstly, the use of a pseudo-random generator implies that they are only computationally secure, but not information-theoretically secure. Secondly, it is impossible to completely remove the bias in the language model itself, and the "provably secure" protocols assumed language models have no distribution bias, but in practice, this is likely not so. Thirdly, the practical implementation (either Arithmetic coding or the Huffman tree) implies that some small distribution shift will be induced. Therefore, perfect security is only theoretically proved, but not in practice. If we can accept this small security leakage, we do not need to focus singularly on the perfect security to start with, as long as we can control it well.

---

> > ### Author Response · Authors · 2024-11-20
> >
> > 1. Regarding the comparisons: Please refer first to the summary response given above. Concerning these more recent methods, we note that most of them adopted the same idea of Ziegler et al., but added certain clever randomization with some additional secret key/PRNG. For example, Meteor essentially extends Ziegler et al.'s work by introducing fresh random seeds; Discop is similar and uses PRNG to randomize the mapping. These clever randomizations help satisfy the strict definition of "security", but they do not fundamentally change the performance. In contrast, we assume the message is already encrypted (therefore distributed i.i.d.), therefore, if we choose $\delta=0$, it is also perfectly secure. It can also be seen in the Meteor and Discop papers, that there was in fact some loss in the embedding utilization (capacity), and the introduction of randomization helps to drive KL-divergence smaller, but still not zero. Our proposed approach, in contrast, allows us to choose a wider range of tradeoff points between utilization and security. Moreover, since these methods only aim at the perfect secure operating points, our proposed approach can be easily incorporated into them to allow near-perfect security in all these techniques.
> >
> > 2. Regarding staganalysis experiments: We appreciate the pointers to the analysis tool, and plan to provide such evaluations if the paper can be accepted. However, we wish to first note that many papers in the literature, e.g., [3] and [4], did not include such analysis. In a way, the KL divergence is the most fundamental measure to evaluate the security (or detection) performance. Data-based analysis tools are going to be weaker evaluators, in the sense that they do not have the full access to the original distribution and the optimized distribution, and the lack of such critical information will clearly make the detection more difficult. Therefore, we expect such an analysis tool to produce results in line with our KL-based evaluation, most likely even more favorable to our proposed method.
> >
> > 3. Regarding the normal sampling of LLMs: We are not completely sure what the reviewer meant here. When the parameter $\delta$ is 0, then the proposed method reduces to normal LLM-generation, i.e., similar to the Meteor method. If the reviewer is questioning why not generate cover-text first, then perform stega-encoding, then we can explain this difference using a ``separation vs joint" processing analogy. If we have access to the underlying generative model, then we can first generate a cover text, then try to embed the secret message by making changes to it; in other words, this cover-text based approach is a two-step procedure, done separately. In contrast, we can achieve steganography in a single shot, by manipulating the generation process directly to embed the secret message without first generating the cover text; this coverless procedure is a joint processing approach. Clearly, a good joint processing will at least be as good as separate processing, since any separate processing procedure is a special case of the joint processing procedure. Prior works by Ziegler, Dai and Cai, indeed showed that the coverless (joint processing) approach is better in terms of security and utilization. Therefore, we chose the more powerful base approach for our study.
> >
> > 4. Regarding the length of B: The receiver does not need the length of B, as they can calculate it using the length of the secret message, which is found during decoding, and equation 10. In our tests, for every hundred files sent, only three to four needed to be repeated, so we would estimate the average repetitions to be around 0.05 at most for the size of the cover text we used.

---

> > > ### Comment · Reviewer_giCB · 2024-11-24
> > >
> > > The author's summary response makes sense to some extent and partially addresses my questions. However, these explanations did not appear in the manuscript, and I don't think this is something that can be explained by space constraints. The author's claim that the newer methodological improvements are minor and therefore not mentioned in the manuscript is, in my opinion, not valid.
> > >
> > > I have always believed that literature research and presentation of relevant work should be a crucial part of a paper, especially for conference papers. From the authors' response, it is not so much that the authors were unfamiliar with the related works since 2020, but more like they intentionally avoided them in the manuscript.
> > >
> > > If the proposed method can be applied to all similar "base" perfect security methods, as the authors claim, then the relevant explanations should be a very important part of the paper. And, the extent to which modifications to the distribution on each method can improve the embedding capacity must also be evaluated.
> > >
> > > Therefore, it might have been easier for me to accept if I could have seen in the modified manuscript the explanation that the author mentioned in this response. But for now, I'll keep my score.

---

> > > > ### Author Response · Authors · 2024-11-25
> > > >
> > > > Thanks for your feedback.
> > > >
> > > > We have revised the manuscript to emphasize the tradeoff perspective between embedding utilization and security. This tradeoff was mentioned in the original submission in a few places but was not emphasized, since we had mostly followed the "near imperceptibility" perspective by Dai and Cai 2019 and Shen et. al. 2020. We have viewed the two perspectives as equivalent mathematically from a constrained optimization point of view.
> > > >
> > > > We were aware of some of the new references in the literature, but after some early discussions where we believed that those contributions were mostly on formalizing the security guarantees, instead of explicitly improving the utilization and security tradeoff, we forgot to include them in the literature review. This was indeed our oversight. We have revised the literature review section in the appendix.
> > > >
> > > > We have not yet programmed and evaluated the approach of incorporating our method into other base methods, but we do plan to do this soon. Intuitively, competitive performances are expected, since they are mostly orthogonal mechanisms, but the reviewer is correct that unless we implement them, we can never be sure what it will lead to.
> > > >
> > > > We again thank the reviewer for his/her constructive comments. We'd greatly appreciate it if the reviewer could look at the revision (lines 60-90) and the appendix (lines 689-702), and provide us some feedback, regardless of whether the reviewer eventually raises the score or not.

---

### Official Review · Reviewer_p2ew · 2024-10-31

**Soundness:** 2
**Presentation:** 3
**Contribution:** 2
**Rating:** 3
**Confidence:** 4

**Summary:**

The author presents a novel steganographic method for large language models (LLMs) that integrates probability truncation and optimized distributions.  This approach aims to embed more secret messages while reducing computational complexity and maintaining undetectability.  The paper's primary contribution lies in addressing tokenization errors associated with the use of tokenizers in existing LLMs for steganography.  Extensive experiments are conducted, employing KL divergence and GPT-4 to evaluate the generated text, with results depicted in clear and comprehensible figures.

**Strengths:**

1. The paper innovatively resolves tokenization errors that occur in existing LLMs used for steganography.
2. The proof for optimizing distribution probabilities is comprehensive and methodologically sound.
3. The graphical representations effectively convey complex information in an accessible manner.

**Weaknesses:**

1. The flow of arithmetic coding steganography in Figure 1 does not include the message extraction operation for the receiver, which is critical for understanding the full process.
2. The experimental section lacks an analysis of how different truncation operations impact embedding and extraction times, which are crucial for assessing practical efficiency.
3. The experiments evaluating GPT-4 only compare different truncation sizes against a baseline, without providing a comparative analysis among these sizes, limiting the understanding of their relative effectiveness.
4. The baseline method used in the experiments is from 2020, and there is no comparison with more recent methods, which may undermine the relevance of the findings.

**Questions:**

1.T he interval obtained from arithmetic coding mapping 0.10111 should be [0.71875, 0.75). Could you clarify this calculation?
2. Could you include the message extraction operation in the flow of arithmetic coding steganography shown in Figure 1 to provide a complete overview?
3. Please consider adding an analysis of the impact of different truncation operations on embedding and extraction times in the experimental section.
4. In the GPT-4 evaluation experiments, only comparisons against the baseline are provided. Can you include a comparative analysis among the different truncation sizes?
5. It would be useful to add examples and requirements for different secret message lengths relative to B bits, as well as the distinctions between correct and incorrect extractions.
6. The baseline method used is from 2020. Could you provide comparisons with the latest methods to contextualize your contributions?

---

> ### Author Response · Authors · 2024-11-20
> **Summary response**
>
> We first thank the reviewer for recognizing the strength of our work. Before proceeding to the item-by-item response, we clarify a few points in a summary below, which were not explained in detail due to space constraints.
>
> 1. Our proposed approach includes the perfectly secure case as a special case. Particularly, note that we assume the secret message has been encrypted. Therefore, the secret message are iid binary (information-theoretic or computational, depending on the encryption process). If we choose the parameter $\delta$ to be zero, i.e., not to increase the embedding utilization by optimizing the distribution, then it is a perfectly secure protocol (almost identical to Meteor).
>
> 2. Our proposed approach essentially provides additional flexibility for system designers to choose a tradeoff point between security and embedding utilization. If in an application, the eavesdropper is resource-limited (or a person paying little attention), then clearly the designer can choose a larger value of $\delta$ to allow more KL-divergence; on the other hand, in other applications, if the security requirement is very high, then $\delta$ should be chosen to be zero, making it completely secure. This additional flexibility is the most important contribution of our work.
>
> 3. The proposed approach can also be easily combined with other techniques, e.g., Discop, Meteor, and ADG. Note that these methods rely on some additional secret keys/randomness, so the overall architecture is somewhat different, however, they all utilize the Ziegler el al. key idea together with some clever randomization techniques. We can optimize the distributions and use the optimized one in Discop, Meteor, or ADG to increase the embedding utilization. Our approach should not be viewed as directly competing with those new er methods but can be viewed as a method to allow further flexibility in them. In other words, our design offers a broader range of design points between a perfect secure solution and the maximum compact embedding solution, on most "base" methods.
>
> 4. Near-perfectly recure steganography is well-accepted in the literature. It had been used widely before coverless steganography became popular. With a covertext, small changes to the covertext always introduce small distribution changes, and near-perfect security has been widely accepted. With more recent coverless steganography, it becomes possible to have "perfect security", such as Discop and Meteor. However, for the coverless setting, "near-imperceptable" steganography was also introduced by Dai and Cai in ACL-19, and it is well-motivated and well-accepted as humans cannot distinguish "perfect security" vs. "near-perfect security".
>
> 5. Some more recent works exist in the literature, however, the improvements over Ziegler's method are minor. For example, both Discop and Meteor are in fact simple variations of Ziegler et al.'s breakthrough idea of using arithmetic coding for steganography. As commented in the Meteor paper, Ziegler's approach is not perfectly secure only due to the reuse of randomness. A minor change to Ziegler's approach will make it perfectly secure: use a random secret key to encrypt the cleartext message (e.g., XOR), then the encrypted secret message can be stega-coded using Ziegler's arithmetic coding approach; the secret key needs to be shared with Bob, but Eve does not have it. Meteor essentially uses this approach, with a simplified arithmetic code that introduces some additional inefficiency (see comments in the Discop paper). Discop uses the random key to instead randomize the Huffman tree (approximately equivalent to fixed-decision arithmetic coding). These newer approaches therefore do not fundamentally impact the performance of Ziegler's, and in fact sometimes they suffer from under-utilization. Ziegler el al.'s work and Dai and Cai's works give the most fundamental methodology, and comparison with them allows a clear picture as to where the strength of our work is, without the distracting factors of randomization techniques in the mix. We can add more comments on the relation if the paper can be accepted.
>
> 6. Even "perfectly secure" protocols such as Discop and Meteor are in fact still not perfectly secure in the strict sense. Firstly, the use of a pseudo-random generator implies that they are only computationally secure, but not information-theoretically secure. Secondly, it is impossible to completely remove the bias in the language model itself, and the "provably secure" protocols assumed language models have no distribution bias, but in practice, this is likely not so. Thirdly, the practical implementation (either Arithmetic coding or the Huffman tree) implies that some small distribution shift will be induced. Therefore, perfect security is only theoretically proved, but not in practice. If we can accept this small security leakage, we do not need to focus singularly on the perfect security to start with, as long as we can control it well.

---

> ### Author Response · Authors · 2024-11-20
>
> 1. Regarding Fig. 1: The decoding part is very similar to the encoding process of arithmetic coding. First, the decoder identifies where the token is located in the probability interval. For example, let the interval be [0.75, 0.875) = [0.110, 0.111), then the receiver knows that the secret bits for the first two tokens are 11. They read the second token and do this process again, and if, for example, the second token is located in the interval [0.78125, 0.8125) = [0.11001, 0.1101),  the receiver knows the secret bits for the first two tokens are ``110". Essentially, one only needs to look at the figure backwards to find the steganography decoding process. Arithmetic coding is a well-known technique in both steganography and data compression, and we intentionally make the introduction here brief. We feel this might be too repetitive to include, but we have indeed revised it on OpenReview to include more explanation for decoding.
>
> 2. Regarding the analysis of truncation method: Our main focus is on the distribution optimization technique, instead of the truncation technique. The latter technique was not introduced by us, but by Dai and Cai ACL-2019. More detailed analysis can be found there. It is clear the more truncation, the faster the process is, however, the impact is relatively small since the bulk of the computation is in the language model. We include this technique here to 1) have a uniform method that includes both ours and the truncation technique, such that the evaluation of the security is more streamlined, and 2) to illustrate that the proposed technique is rather flexible and fundamental, and can be compatible with this well-known technique. The coding time is dominated by the LLM computation, and the differences in other components are rather negligible in this sense.
>
> 3. Regarding Fig. 5: we have revised the figure and will upload an updated version.
>
> 4. Regarding the choice of the baselines: Please refer first to the general response given above. Reviewer 3 provided several more recent works.  Concerning these more recent methods, we note that most of them adopted the same idea of Ziegler et al., but added certain clever randomization with some additional secret key/PRNG. For example, Meteor essentially extends Ziegler et al.'s work by introducing fresh random seeds; Discop is similar and uses PRNG to randomize the mapping. These clever randomizations help satisfy the strict definition of "security", but they do not fundamentally change the performance in terms of utilization and security. In contrast, we assume the message is already encrypted (therefore distributed i.i.d. Bernoulli), therefore, if we choose $\delta=0$, it is also perfectly secure. It can also be seen in the Meteor and Discop papers, that there was in fact some loss in the embedding utilization (capacity), and the introduction of randomization helps to drive KL-divergence smaller, but still not zero. Our proposed approach, in contrast, allows us to choose a wider range of tradeoff points between utilization and security. Moreover, since these methods only aim at the perfect secure operating points, our proposed approach can be easily incorporated into them to allow near-perfect security in all these techniques.
>
> 5. Regarding the interval in the example: This was a typo introduced inadvertently when we attempts to make the example more readable. Indeed this should be [0.71875, 0.75). Thanks for pointing this out.
>
> 6. Regarding more examples: More examples can be found in the appendix. We can add further examples if the paper is eventually accepted.

---

### Official Review · Reviewer_oWxW · 2024-11-02

**Soundness:** 2
**Presentation:** 3
**Contribution:** 2
**Rating:** 3
**Confidence:** 5

**Summary:**

The paper suggest to distort little bit the probability distribution of LLM in generative steganography to increase the capacity.

**Strengths:**

* The theory of the paper is decent.
* It is refreshing to see terminology used in steganographic literature.
* I have never thought about issues with tokenization, as I have thought about it about unique. But it is not. This is a big deal jeopardizing a lot of prior art.

**Weaknesses:**

There are few concerns I have with the work.
* Steganography always thrived to be undetectable. One of the big hallmarks of generative steganography is that if done right, it is actually undetectable. Why would I want to trade off undetectability for a higher capacity? This is very important when one considers square-root law discovered by Andrew Ker [1]. This says that the message embedded by an imperfect steganographic system, which authors have introduced has to decrease faster than square-root of length of the cover, otherwise the system will be detectable. With respect to this, the proposed system have sense only for one-off message where the steganography is not repeated (see works by Andrew Ker on pooled steganalysis [2]). With respect to this, I think that for longer messages and / or repeated communication the improvement in the capacity will be small, otherwise the steganographer will be detectable in the long run. Or the other way around, the method proposed in the paper makes sense when the steganographer sends only one short message. I am not sure this is of practical interest.
* The second flaw of the paper is that it distorts probability of individual tokens. In some sense, the theory therefore assumes IID model and does not take interactions between tokens and the cumulative effects into the consideration. This is not necessarily bad, as one of the big hallmarks of cover-modification steganography was derived under exactly the same conditions, but then it took more than a decade to derive a principled method utilizing this system [4]. But this point should be made clear and explicit. Again, with respect to the above claim.
* My third objection is the evaluation. It seems to me that it makes the Eve (warden, steganalyst) just plain weak. Under Kerckhoffs' principle, she has all the knowledge about the steganographic channel. Why she cannot train a statistical detector, for example by finetuning some LLM, and relies instead on some instructed LLM? This seems just plain wrong to me.
* Discop [5] on page 4 discusses practical issues of arithmetic coding and argues that unless the length of the message is very long, there will be a detectable distortion (see Table II of paper [5]). Your scheme further increases this distortion.


[1] Ker, Andrew D. "The square root law in stegosystems with imperfect information." Information Hiding: 12th International Conference, IH 2010, Calgary, AB, Canada, June 28-30, 2010, Revised Selected Papers 12. Springer Berlin Heidelberg, 2010.

[2] Ker, Andrew D. "Batch steganography and pooled steganalysis." International Workshop on Information Hiding. Berlin, Heidelberg: Springer Berlin Heidelberg, 2006.

[3] Fridrich, Jessica, and Tomas Filler. "Practical methods for minimizing embedding impact in steganography." Security, Steganography, and Watermarking of Multimedia Contents IX. Vol. 6505. SPIE, 2007.

[4] Bernard, Solène, et al. "Optimizing additive approximations of non-additive distortion functions." Proceedings of the 2021 ACM Workshop on Information Hiding and Multimedia Security. 2021.

[5] Ding, Jinyang, et al. "Discop: Provably secure steganography in practice based on" distribution copies"." 2023 IEEE Symposium on Security and Privacy (SP). IEEE, 2023.

**Questions:**

I would like to ask authors, if they can comment the weaknesses I have mentioned above.

---

> ### Author Response · Authors · 2024-11-20
> **Summary response**
>
> We first thank the reviewer for recognizing the strength of our work. Before proceeding to the item-by-item response, we clarify a few points in a summary below, which were not explained in detail due to space constraints.
>
> 1. Our proposed approach includes the perfectly secure case as a special case. Particularly, note that we assume the secret message has been encrypted. Therefore, the secret message are iid binary (information-theoretic or computational, depending on the encryption process). If we choose the parameter $\delta$ to be zero, i.e., not to increase the embedding utilization by optimizing the distribution, then it is a perfectly secure protocol (almost identical to Meteor).
>
> 2. Our proposed approach essentially provides additional flexibility for system designers to choose a tradeoff point between security and embedding utilization. If in an application, the eavesdropper is resource-limited (or a person paying little attention), then clearly the designer can choose a larger value of $\delta$ to allow more KL-divergence; on the other hand, in other applications, if the security requirement is very high, then $\delta$ should be chosen to be zero, making it completely secure. This additional flexibility is the most important contribution of our work.
>
> 3. The proposed approach can also be easily combined with other techniques, e.g., Discop, Meteor, and ADG. Note that these methods rely on some additional secret keys/randomness, so the overall architecture is somewhat different, however, they all utilize the Ziegler el al. key idea together with some clever randomization techniques. We can optimize the distributions and use the optimized one in Discop, Meteor, or ADG to increase the embedding utilization. Our approach should not be viewed as directly competing with those new er methods but can be viewed as a method to allow further flexibility in them. In other words, our design offers a broader range of design points between a perfect secure solution and the maximum compact embedding solution, on most "base" methods.
>
> 4. Near-perfectly recure steganography is well-accepted in the literature. It had been used widely before coverless steganography became popular. With a covertext, small changes to the covertext always introduce small distribution changes, and near-perfect security has been widely accepted. With more recent coverless steganography, it becomes possible to have "perfect security", such as Discop and Meteor. However, for the coverless setting, "near-imperceptable" steganography was also introduced by Dai and Cai in ACL-19, and it is well-motivated and well-accepted as humans cannot distinguish "perfect security" vs. "near-perfect security".
>
> 5. Some more recent works exist in the literature, however, the improvements over Ziegler's method are minor. For example, both Discop and Meteor are in fact simple variations of Ziegler et al.'s breakthrough idea of using arithmetic coding for steganography. As commented in the Meteor paper, Ziegler's approach is not perfectly secure only due to the reuse of randomness. A minor change to Ziegler's approach will make it perfectly secure: use a random secret key to encrypt the cleartext message (e.g., XOR), then the encrypted secret message can be stega-coded using Ziegler's arithmetic coding approach; the secret key needs to be shared with Bob, but Eve does not have it. Meteor essentially uses this approach, with a simplified arithmetic code that introduces some additional inefficiency (see comments in the Discop paper). Discop uses the random key to instead randomize the Huffman tree (approximately equivalent to fixed-decision arithmetic coding). These newer approaches therefore do not fundamentally impact the performance of Ziegler's, and in fact sometimes they suffer from under-utilization. Ziegler el al.'s work and Dai and Cai's works give the most fundamental methodology, and comparison with them allows a clear picture as to where the strength of our work is, without the distracting factors of randomization techniques in the mix. We can add more comments on the relation if the paper can be accepted.
>
> 6. Even "perfectly secure" protocols such as Discop and Meteor are in fact still not perfectly secure in the strict sense. Firstly, the use of a pseudo-random generator implies that they are only computationally secure, but not information-theoretically secure. Secondly, it is impossible to completely remove the bias in the language model itself, and the "provably secure" protocols assumed language models have no distribution bias, but in practice, this is likely not so. Thirdly, the practical implementation (either Arithmetic coding or the Huffman tree) implies that some small distribution shift will be induced. Therefore, perfect security is only theoretically proved, but not in practice. If we can accept this small security leakage, we do not need to focus singularly on the perfect security to start with, as long as we can control it well.

---

> > ### Author Response · Authors · 2024-11-20
> >
> > 1. Regarding the detectability and square-root law: Indeed, the square-root law [1] is an asymptotic result, which essentially says when the secret message is infinite long, any difference in the distribution can be detected. This is not surprising, but has little relevance to the practical situation we feel is most relevant. In practice, the secret message will not be exceedingly long. Moreover, this stego-text is often only a part of the overall text. For example, in a more realistic situation, the first paragraph can be meaningful text that Alice wishes to convey, then a certain secret message can be embedded in a few sentences of the second paragraph using our approach, and the normal text follows after that. In such cases, it is clearly very important to squeeze the secret message into the short space given. Considering extremely long secret messages eases the theory development in [1], but in practice, we feel that is rarely the case. We can make this point more explicit in a revision if the reviewer feels it is important.
> >
> > 2. Regarding the individual token processing: We are fully aware of this issue. We first note that in Dai and Cai ACL-19, they showed that the sequence-wise KL divergence is essentially the sum of individual KL divergence. Therefore, we can be assured that as long as individual KL is kept low, the method will not break the overall KL-divergence in a catastrophic manner. Secondly, this individual-token approach is widely used in LLM-based steganography, due to the practical computation consideration with LLMs. Let us suppose one wishes to plan ahead here, then the strategy would have to essentially combine the probability distribution of the next several tokens to form a joint distribution to choose the optimized distribution, but this would entail running language models many times. We can potentially approximate it, using methods similar to a beam search. However, such a strategy is not realistic in LLM-based steganography, since computation is an important consideration, and LLM is already quite computation-heavy. Thirdly, we introduced some considerations in the design to alleviate such issues, e.g., embedding in high-entropy positions only, and $\delta$ value depending on the entropy. We would be happy to include some discussion on this issue in the revision.
> >
> > 3. Regarding the evaluation: We have to disagree with this comment. We evaluate the overall KL divergence between the original distribution and the optimized one. According to data processing inequality of the KL divergence, any subsequent processing will not make it easier to distinguish. The KL divergence can be viewed as the "ultimate test" if one wishes to distinguish between two distributions in this sense. The best a trained detector can do in this situation will be similar to learning how to calculate the KL divergence, and perform certain thresholding. Another reviewer suggests analysis tools for such evaluation, and we plan to incorporate such evaluation if the paper is accepted. However note that all data-based analysis tools are likely weaker evaluators, in the sense that they do not have the full access to the original distribution and the optimized distribution, and the lack of such critical information will clearly make the detection more difficult. Therefore, we expect such an analysis tool to produce results in line with our KL-based evaluation, most likely even more favorable to our proposed method. We further note that our setting goes beyond merely "detection": we allow other tradeoff points between utilization and security in a general manner, and any trained models for detection can only provide classification instead of meaningful evaluation of the tradeoff.
> >
> > 4. Regarding the issue of arithmetic coding: The authors of [5] speculated that the non-zero KL divergence of ADG is due to the arithmetic coding. However, we believe this was not the right explanation. Instead, the ADG paper takes a different strategy from arithmetic coding, and attempts to make the grouping have a uniform distribution. The authors of ADG even mentioned in their paper that this strategy will induce KL loss if the starting distribution is such that it is not possible to make the distribution perfectly uniform (section 4.1).

---

> > ### Comment · Reviewer_oWxW · 2024-11-20
> > **Official response**
> >
> > I gladly agree with you that theoretically secure methods were in past frequently broken, because the assumptions were not realistic.
> >
> > So let's now assume a realistic problem that I need to send a message of a certain size. In the text generative steganography, I have multiple options:
> >
> > 1. I can increase the payload at the expense of the distortion.
> > 2. I can generate longer text.
> > 3. I can split the message into more messages.
> >
> > In your paper, you propose version 1, but I think that version 2 and 3 are both valid and obvious strategies. What is the advantage of the option 1. Should not you experimentally convince me that 1 is more secure?
> >
> > I also confess that my preferred definition of steganography is based on statistically detectability, which is usually estimated by error of a trained detector. Therefore when showing security of the scheme, I would prefer an error of a decent detector, such as fine-tuned GPT. This would be more interpretable than KL-divergence and would allow to well understand the security trade-offs.

---

> > > ### Author Response · Authors · 2024-11-20
> > >
> > > Thanks for the thoughts and questions.
> > >
> > > We believe engineering is not one-size-fits-all. Approach 1 as outlined above is important when the generated stego-text cannot be too long. For example, perhaps the stego-text needs to be in a single tweet, instead of a full-page letter or in many tweets. In such a length-constrained setting, we cannot use approach 2 or 3. As engineers, we aim to offer more flexible design choices instead of forcing one solution. Moreover, our approach in fact includes approaches 2 or 3, since setting $\delta=0$ can recover those.
> > >
> > > To make things even more concrete, let us consider two scenarios: A) There is an all-powerful Eve with unlimited computation resources, B) Eve only has the intelligence of a 5-year-old. If we know that we are dealing with case (B), don't we want to take advantage of it by embedding more information in shorter stego-text? This would allow less communication overall, or more hidden information in the same passage. Our proposed method provides such a choice. If we instead know it is actually case (A), our approach can also handle it by adjusting the parameter to 0. Most existing methods only focus on (A), but ignore (B) or similar settings (except Dai and Cai). In other words, perceptibility depends on the ability of Eve, and the choice of $\delta$ allows different operating points for different Eve's with different capabilities.
> > >
> > > To summarize, the parameter $\delta$ is a control parameter that would allow different tradeoffs between embedding utilization and security in different engineering scenarios. The proposed approach either includes previous approaches as a special case, or can be incorporated easily into them to provide a more flexible design.
> > >
> > > Using LLM-based detector is indeed possible and we are willing to consider adding such a test if the paper is accepted, but it is clear to us that statistical detector will be weaker. This is because they need to "learn" the statistics, yet the KL value in fact uses the "ground-truth" statistics directly. It is unrealistic to expect a learned discriminator to do better than one that already knows the ground truth. We feel that the issue may be that it is difficult to interpret the KL value. However, recall that we are dealing with a design that offers a wide range of KL values, and the system designer should choose the right value for their individual application. An LLM-based detector cannot handle such varying requirements. Moreover, suppose we set $\delta$ to be a very small value, say 10^-8, and show that an LLM-based detector cannot do better than random guesses, will that be considered convincing? Actually, in this special case, our method is almost just Meteor, and we will be simply reevaluating whether Meteor is secure or not, but this does not help with the evaluation of the general tradeoff.

---

> > > > ### Comment · Reviewer_oWxW · 2024-11-21
> > > >
> > > > I apologise, but I am not convinced by the arguments. For me, steganography is all about being undetectable, because there is not sense of using steganography to be detectable.
> > > >
> > > > It is long known in steganography by cover modification that if Alice know Eve's detector, then the solution is trivial, see eg. [1,2] below. Alice can for example use variant of rejection sampler, where she would repeatedly produce stego objects and selected the one, which is least detectable.
> > > >
> > > > The point of the study I was suggesting was that you fix the payload and then show, than strategies 2 and 3 are worse than 1. It other words, shorter message with different distribution might be better than longer message. The point is, that it is certainly possible your approach will be better, since longer message gives Eve more statistics to detect deviations to cover distribution (the above mentioned square root law).  But I want to see that.
> > > >
> > > > The reason why I prefer learned detector over KL-divergence of ground truth model is that a) I do not know, how to interpret the number provided by KL-divergence  and b) Eve has to learn the statistic of cover distribution as Alice does [3], therefore the setting is I think quite realistic.
> > > >
> > > >
> > > > [1] Tang, Weixuan, et al. "CNN-based adversarial embedding for image steganography." IEEE Transactions on Information Forensics and Security 14.8 (2019): 2074-2087.
> > > > [2] Pevny, Tomas, and Andrew D. Ker. "Exploring non-additive distortion in steganography." Proceedings of the 6th ACM Workshop on Information Hiding and Multimedia Security. 2018.
> > > > [3] Böhme, Rainer. "An epistemological approach to steganography." International Workshop on Information Hiding. Berlin, Heidelberg: Springer Berlin Heidelberg, 2009.

---

> > > > > ### Author Response · Authors · 2024-11-22
> > > > >
> > > > > Thanks for the feedback.
> > > > >
> > > > > The result that "Alice knows Eve's detector then rejection sampling can be used" requires that Alice knows the detector with full knowledge. However, as we described above, if Alice only knows that Eve is a weaker detector, but does not know with full knowledge (the mechanism) how Eve is performing the detection, then that result breaks down. I'm sure the reviewer is familiar with the differential privacy literature, where the "epsilon" and "delta" parameters are used to reflect similar considerations. More generally, this class of guarantees is referred to as "relative security".
> > > > >
> > > > > Regarding the different approaches outlined by the reviewer, they are in fact solutions suitable for different scenarios, and shouldn't be compared head-to-head. An analogy is that in Google/IOS Map, we have the option of choosing "speed priority", "gas-mileage priority", or "distance priority". Do we value them and say one is better than the other? One choice can be better than the others in one scenario, while vice versa in another scenario. In steganography, approach 1 is better if embedding utilization is prioritized (e.g., heavily constrained by the stego-text length for a fixed payload), and approach 2/3 is better if security is prioritized (less stringent constrained by the stego-text length for a fixed payload). Our proposed approach allows us to make this adjustment using the parameter $\delta$.
> > > > >
> > > > > We can understand the reviewer's desire to see the performance with a detector based on learning. We are willing to try it. Nevertheless, the best one can say with such a trained detector is perhaps to find a KL threshold under which the difference is not detectable in the test environment. However, this might not be as useful as the reviewer would expect, since the usage of steganography primitive can be much more flexible, and the thresholds can be different, e.g., it may depend on the type of context (sports news vs. story-telling). We would rather leave that as a designing lever for application developers, instead of stating "0.2 is a good choice", which might be quite misleading. Using again the differential privacy analogy, we don't see people claiming "delta<0.01 guarantees sufficient privacy".
> > > > >
> > > > > It might be hard to convince the reviewer through this channel, but to help us understand the issue and improve, we would appreciate candid answers from the reviewer for the following two questions:
> > > > > 1. In the reviewer's mind, if Alice and Bob know the eavesdropper Eve only has the intelligence of a 5-year-old but does not know exactly how Eve behaves, does the argument make sense that they should take advantage by relaxing the security requirement in some way? I.e., do we want to make it imperceptible to that weaker eavesdropper?
> > > > > 2. Will it be helpful if we choose a title that uses "relatively secure steganography", instead of "near-imperceptable"?
> > > > >
> > > > > Thanks again for the help.

---

> > > > > > ### Comment · Reviewer_oWxW · 2024-11-22
> > > > > >
> > > > > > To be honest, I do not believe in the application. I simply cannot imagine where I would be forced to use steganography (not mere data hiding) and I would increase detectability of my hiding. When I need to communicate message with given payload, I was suggesting to compare your proposal to different strategies to increase the capacity. I have explained that the other strategies also increase statistical detectability due to square-root law. But this is fair point. I honestly do not like your argument that these simple strategies I have suggested do not apply to your scenario.
> > > > > >
> > > > > > Also, how is it likely that strategic steganographer in important application would be facing Eve with a mental capability of 5 years old child. How many kids at that age knows steganography after all?

---

> > > > > > > ### Author Response · Authors · 2024-12-01
> > > > > > >
> > > > > > > The "5-year intelligence" analogy is for informal interpretation. Formally, we can state our motivation as follows: "In many use scenarios, the security requirement in steganography can in fact be relaxed: 1) when Eve is computation-bounded (e.g., in a mobile device), 2) when Eve is delay-constrained (e.g., in streaming processing or time-sensitive applications), or 3) under societal constraint (e.g., censorship under constitutional right protection). In such cases, Eve can be modeled as a weak detector, and correspondingly the steganography security requirement can be relaxed to take advantage of the situation."
> > > > > > >
> > > > > > > It would be difficult to convince the reviewer to "believe" the use cases and formulation, but we note that even the original perfect-secure steganography is only a mathematical model that abstracts some practical applications. There are plenty of things to criticize and dislike about that model (e.g., not taking into account the language model distribution bias and the computation resource restriction at Alice/Bob), but that did not stop people from using/researching/publishing it.
> > > > > > >
> > > > > > > We feel that we have addressed all the reviewer's concerns from a technical and objective angle. Though we truly appreciate the reviewer's feedback, we also feel subjective comments like "don't believe" or "don't like" are not the most helpful in a technical review.

---

### Author Response · Authors · 2024-11-22

We appreciate the input from Reviewer oWxW, but would also like to hear thoughts from the other reviewers. Thanks for your help in improving our work.

---

### Meta-Review · Area_Chair_Hmwu · 2024-12-17

**Metareview:**

This paper proposes to distort the probability distribution of LLM in generative steganography. In particular, the distortion of the probability distribution is done by integrating probability truncation and optimized distributions. This approach aims to embed more secret messages while reducing computational complexity and maintaining undetectability. The paper's primary contribution lies in addressing tokenization errors associated with the use of tokenizers in existing LLMs for steganography. Extensive experiments are conducted, employing KL divergence and GPT-4 to evaluate the generated text.

However, the reviewers remain unconvinced about the methodology proposed. Among other issues (several that I will not restate), the reviewers criticized the following.

1) In particular, the evaluation metric for the security part is the KL-divergence, which cannot be related to actual amount of security in practice.
2) Steganography is meant to be undetectable. It is not clear why one would want to trade off undetectability with capacity.
3) Baseline methods considered are old, before 2020.

These were not adequately resolved even after the discussion period. Hence, this work, while promising, is premature and not ready for publication.

**Additional Comments On Reviewer Discussion:**

The authors responded to the reviewers. Their replies are well considered and thoughtful. However, the reviewers maintained that the deficiencies highlighted above remain even after several exchanges. The practicality of the formulation (e.g., IID tokens), in particular, is questioned.

---

### Decision · Program_Chairs · 2025-01-22

Reject